# Long read sequencing reveals poxvirus evolution through rapid homogenization of gene arrays

Thomas A Sasani[†], Kelsey R Cone[†], Aaron R Quinlan[‡]*, Nels C Elde[‡]*

Department of Human Genetics, University of Utah, Salt Lake, United States

**Abstract** Poxvirus adaptation can involve combinations of recombination-driven gene copy number variation and beneficial single nucleotide variants (SNVs) at the same loci. How these distinct mechanisms of genetic diversification might simultaneously facilitate adaptation to host immune defenses is unknown. We performed experimental evolution with vaccinia virus populations harboring a SNV in a gene actively undergoing copy number amplification. Using long sequencing reads from the Oxford Nanopore Technologies platform, we phased SNVs within large gene copy arrays for the first time. Our analysis uncovered a mechanism of adaptive SNV homogenization reminiscent of gene conversion, which is actively driven by selection. This study reveals a new mechanism for the fluid gain of beneficial mutations in genetic regions undergoing active recombination in viruses and illustrates the value of long read sequencing technologies for investigating complex genome dynamics in diverse biological systems.

DOI: https://doi.org/10.7554/eLife.35453.001

*For correspondence:
aaronquinlan@gmail.com (ARQ);
nelde@genetics.utah.edu (NCE)

[†]These authors contributed equally to this work
[‡]These authors also contributed equally to this work

## Introduction

Gene duplication is long recognized as a potential source of genetic innovation (*Ohno, 1970*). Following duplication events, the resulting stretches of homologous sequence can promote recombination between gene copies. Gene conversion, the nonreciprocal transfer of sequence between homologous genetic regions, is one outcome of recombination evident in diverse eukaryotes (*Brown et al., 1972*; *Semple and Wolfe, 1999*; *Drouin, 2002*; *Rozen et al., 2003*; *Ezawa et al., 2006*; *Chen et al., 2007*), as well as in bacterial and archaeal genomes (*Santoyo and Romero, 2005*; *Soppa, 2011*). Gene conversion can result in a high degree of identity among duplicated gene copies, as in ribosomal RNA gene arrays (*Liao, 1999*; *Eickbush and Eickbush, 2007*). However, in other cases, such as the human leukocyte antigen gene family (*Zangenberg et al., 1995*) or the transmembrane protein gene cassettes of some pathogenic bacteria (*Santoyo and Romero, 2005*), gene conversion can also generate sequence diversity. Because multiple gene copies create more targets for mutation, variants that arise within individual copies can be efficiently spread or eliminated through gene conversion (*Mano and Innan, 2008*; *Ellison and Bachtrog, 2015*).

Although recombination might influence genetic variation in populations on very short time scales, many studies to date have used phylogenetic analyses in relatively slow-evolving populations to infer outcomes of gene conversion, including cases of concerted evolution, where similarity between genes of a gene family within a species exceeds the similarity of orthologous genes between species (*Chen et al., 2007*; *Ohta, 2010*). Extensive studies in yeast and bacteria (reviewed in *Petes and Hill, 1988*; *Perkins, 1992*; *Haber, 2000*; *Santoyo and Romero, 2005*), and some recent work in viruses (*Hughes, 2004*; *Ba Abdullah et al., 2017*), consider gene conversion on shorter time scales. In order to expand our understanding of how recombination might influence virus variation during the course of adaptation, we focused on large DNA viruses, in which rapidly

evolving populations can simultaneously harbor both adaptive gene copy number variation and beneficial single nucleotide variants (SNVs) at the same locus.

Poxviruses are an intriguing system to study mechanisms of rapid adaptation, as they possess high rates of recombination (*Ball, 1987*; *Evans et al., 1988*; *Spyropoulos et al., 1988*; *Merchlinsky, 1989*) that lead to the recurrent emergence of tandem gene duplications (*Slabaugh et al., 1989*; *Elde et al., 2012*; *Brennan et al., 2014*; *Erlandson et al., 2014*; *Cone et al., 2017*). The poxvirus DNA polymerase gene encodes both replicase and recombinase activities, reflecting a tight coupling of these essential functions for virus replication (*Colinas et al., 1990*; *Willer et al., 1999*; *Hamilton and Evans, 2005*). Polymerase-associated recombination may underlie the rapid appearance of gene copy number variation (CNV), which was proposed as a potentially widespread mechanism of vaccinia virus adaptation in response to strong selective pressure (*Elde et al., 2012*; *Cone et al., 2017*). In these studies, recurrent duplications of the K3L gene, which encodes a weak inhibitor of the human innate immune factor Protein Kinase R (PKR; *Davies et al., 1992*), were identified following serial infections of human cells with a vaccinia strain lacking a strong PKR inhibitor encoded by the E3L gene (ΔE3L; *Chang et al., 1992*; *Beattie et al., 1995*). In addition to copy number amplification, a beneficial single nucleotide variant arose in the K3L gene in some populations, resulting in a His47Arg amino acid change (K3L$^{His47Arg}$), which encodes enhanced inhibition of PKR activity and aids virus replication (*Kawagishi-Kobayashi et al., 1997*; *Elde et al., 2012*). However, the mechanisms by which these distinct processes of adaptation might synergize or compete in virus populations during virus evolution remain unknown.

To investigate how heterogeneous virus populations adapt to cellular defenses, we performed courses of experimental evolution with a vaccinia virus population containing both K3L CNV and the K3L$^{His47Arg}$ SNV (*Elde et al., 2012*). To overcome the challenge of genotyping point mutations in repetitive arrays from evolving populations, we sequenced virus genomes with the Oxford Nanopore Technologies (ONT) MinION platform. Sequencing extremely long DNA molecules allowed us to develop an integrated pipeline to analyze CNV at single-genome scale. Long sequencing reads, which can completely span tandem arrays of K3L duplications (reads measuring up to 99 kbp and comprising up to 16 K3L copies in this study), provided a means for tracking the K3L$^{His47Arg}$ mutation as it spread through K3L gene arrays within an evolving virus population. Altering conditions in this experimental system allowed us to assess the impact of selection and recombination on K3L$^{His47Arg}$ variant accumulation within K3L arrays. These analyses of variant dynamics reveal a mechanism of virus adaptation involving genetic homogenization and demonstrate how long read sequencing can facilitate studies of recombination-driven genome evolution.

## Results

### Rapid accumulation of a single nucleotide variant following gene copy number amplification

In previous work, we collected a virus population adapted over ten serial infections that contained gene copy number amplifications of K3L, and the beneficial K3L$^{His47Arg}$ point mutation at a frequency of roughly 0.1 in the population (*Figure 1A and B*, up to passage 10; *Elde et al., 2012*). To study the fate of the K3L$^{His47Arg}$ variant among repetitive arrays of K3L, we performed ten additional serial infection passages (P11-P20) in human cells. Comparative replication in human cells showed that virus titers remained well above parent (ΔE3L) levels through P20 (*Figure 1A*). While there was no major gain in replication between P5 and P20, the resolution of titer measurements may limit detection of more subtle increases in virus replication. A clear increase in replication around P5 coincided with the emergence of K3L CNV within virus genomes (*Elde et al., 2012*), and gene copy number increases appeared to stabilize by P10 (*Figure 1B*). Notably, the K3L$^{His47Arg}$ SNV, though apparently stable at an estimated frequency of 0.1 in the population from P5 through P10 (*Elde et al., 2012*), accumulated to near fixation between P10 and P20 (maximum frequency of approximately 0.9; *Figure 1B*). Thus, despite a plateau in replication across later experimental passages, the accumulation of the beneficial point mutation suggests that there is selection for the K3L$^{His47Arg}$ variant in the heterogeneous virus population.

To determine how changes in K3L copy number and the K3L$^{His47Arg}$ mutation might individually contribute to virus replication in heterogeneous populations, we isolated distinct variants from single

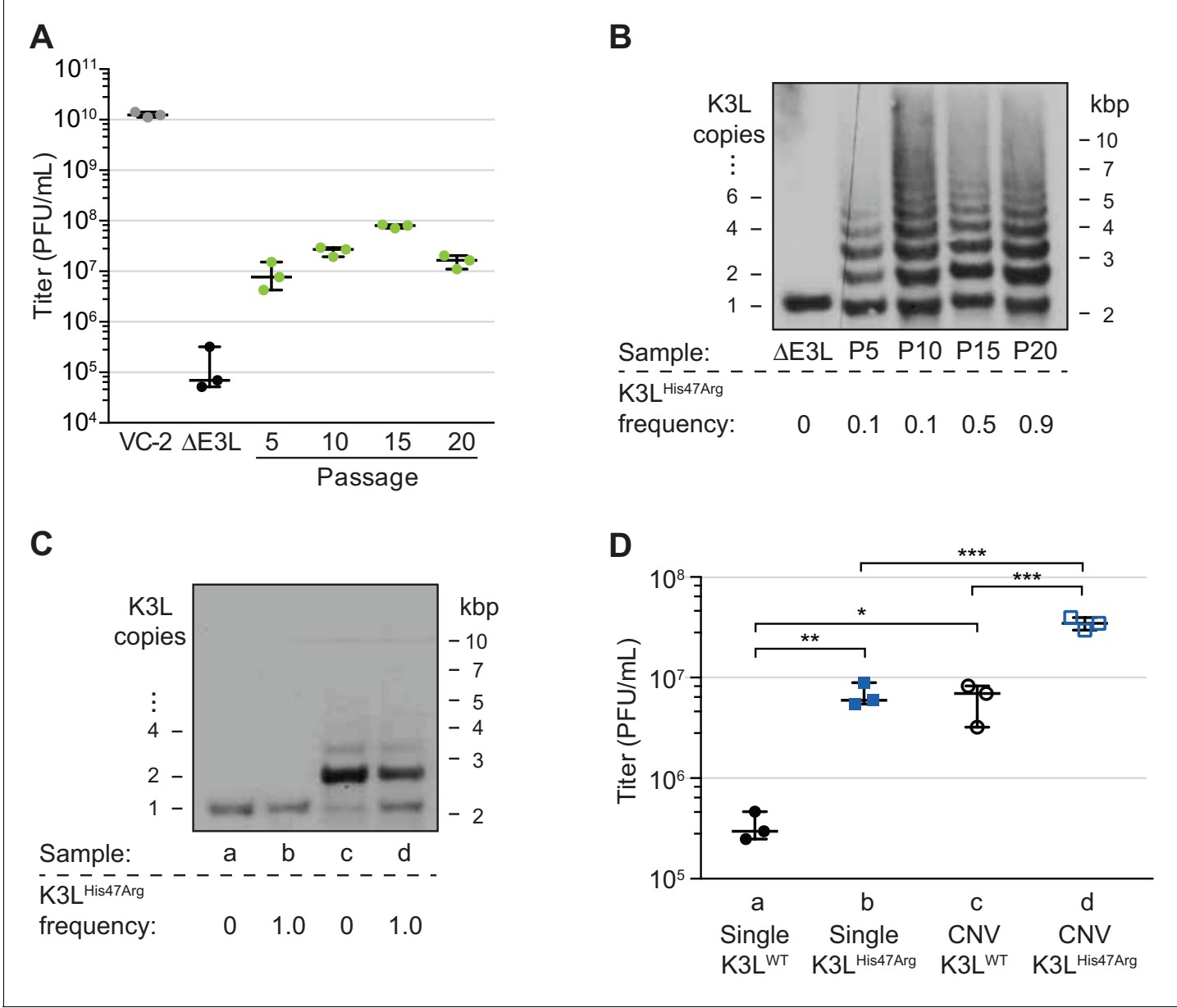

**Figure 1.** A single nucleotide variant accumulates following increases in K3L copy number. (**A**) Following 20 serial infections of the ΔE3L strain (MOI 0.1 for 48 hr) in HeLa cells (see Materials and methods for further details), replication was measured in triplicate in HeLa cells for every fifth passage, and compared to wild-type (VC-2) or parent (ΔE3L) virus. (**B, C**) Digested viral DNA from every 5th passage (**B**) and four plaque-purified clones (**C**) were probed with a K3L-specific probe by Southern blot analysis. Number of K3L copies (left) and size in kbp (right) are shown. K3L$^{His47Arg}$ allele frequency for each population (shown below) was estimated by PCR and Sanger sequencing of viral DNA. (**D**) Replication of plaque purified clones from (**C**) was measured in HeLa (**D**) or BHK (*Figure 1—figure supplement 1*) cells in triplicate. Statistical analysis was performed to compare the means of populations *b* or *c* relative to *a*, or between the means of populations *b* or *c* relative to *d* by one-way ANOVA followed by Dunnett's multiple comparison test. *p<0.05, **p<0.01, ***p<0.005. K3L$^{His47Arg}$ and E9L$^{Glu495Gly}$ population-level allele frequencies estimated from Illumina MiSeq reads are shown in *Figure 1—figure supplement 2*. Replication of clone *a* compared to ΔE3L is shown in *Figure 1—figure supplement 3*. All titers were measured multiple times in BHK cells by plaque assay, shown with median and 95% confidence intervals.

DOI: https://doi.org/10.7554/eLife.35453.002

The following source data and figure supplements are available for figure 1:

**Source data 1.** Data used to generate *Figure 1A*.
DOI: https://doi.org/10.7554/eLife.35453.006

**Source data 2.** Data used to generate *Figure 1D*.
DOI: https://doi.org/10.7554/eLife.35453.007

*Figure 1 continued on next page*

*Figure 1 continued*

**Source data 3.** Statistics for *Figure 1D*, One-way ANOVA followed by Dunnett's multiple comparison test.
DOI: https://doi.org/10.7554/eLife.35453.008

**Figure supplement 1.** K3L[His47Arg] and K3L CNV are non-adaptive in the permissive BHK cell line.
DOI: https://doi.org/10.7554/eLife.35453.003

**Figure supplement 1—source data 1.** Data used to generate *Figure 1—figure supplement 1*.
DOI: https://doi.org/10.7554/eLife.35453.009

**Figure supplement 1—source data 2.** Statistics for *Figure 1—figure supplement 1*, One-way ANOVA followed by Dunnett's multiple comparison test.
DOI: https://doi.org/10.7554/eLife.35453.010

**Figure supplement 2.** Allele frequencies of the two high-frequency SNVs identified in vaccinia populations.
DOI: https://doi.org/10.7554/eLife.35453.004

**Figure supplement 2—source data 3.** Data used to generate *Figure 1—figure supplement 2*.
DOI: https://doi.org/10.7554/eLife.35453.011

**Figure supplement 3.** The E9L[Glu495Gly] variant does not contribute to virus replication.
DOI: https://doi.org/10.7554/eLife.35453.005

**Figure supplement 3—source data 4.** Data used to generate *Figure 1—figure supplement 3*.
DOI: https://doi.org/10.7554/eLife.35453.012

**Figure supplement 3—source data 5.** Statistics for *Figure 1—figure supplement 3*, unpaired 2-tailed t test with Welch's correction.
DOI: https://doi.org/10.7554/eLife.35453.013

virus clones. Following plaque purification, we obtained viruses containing a single copy of the K3L gene, either with or without the K3L[His47Arg] variant (*Figure 1C*). We also obtained plaque purified clones containing K3L CNV that were homogeneous for either wild-type K3L (K3L[WT]) or K3L[His47Arg]. While viruses with CNV were not clonal due to recurrent recombination between multicopy genomes, they possessed nearly uniform K3L copy number, collapsing under relaxed selection to contain mainly 2 copies of K3L following plaque purification (ranging from 1 to 5; *Figure 1C*), consistent with observations from our previous study (*Elde et al., 2012*). As a result, these plaque purified clones do not represent the total diversity of copy number observed in the passaged virus populations, and the individual contributions of each K3L[WT] or K3L[His47Arg] copy to virus replication are difficult to pinpoint given the heterogeneity of the clones. However, these populations do provide a useful tool to approximate the replication of viruses containing distinct genetic changes. Comparing the ability of these viruses to replicate in human cells revealed that either the K3L[His47Arg] variant or K3L CNV is sufficient for a replication gain, and the combination of the two genetic changes increases replication more than either one alone (*Figure 1D*). In contrast, in a permissive hamster cell line, there was less than a 2-fold difference in titer between any of these four viruses (*Figure 1—figure supplement 1*). These results are consistent with earlier reports showing that K3L containing the His47Arg variant is a more potent inhibitor of human PKR than wild-type K3L (*Kawagishi-Kobayashi et al., 1997*), and that CNV was sufficient for overexpression of the K3L protein, which increased virus replication in human cells (*Elde et al., 2012*). However, the K3L[His47Arg] variant only reached high frequency in our experimental population following increases in K3L copy number. This suggests that copy number affected K3L[His47Arg] accumulation, which could occur through changes to selection for the variant, recombination rate, replication rate, or some combination of these processes.

In addition to K3L, variation at other loci could influence vaccinia adaptation during experimental evolution. Therefore, we assessed the full complement of single nucleotide variants by sequencing virus genomes from passages 10, 15, and 20 using the Illumina MiSeq platform. Apart from the K3L[His47Arg] variant, only one other SNV was identified above an allele frequency of 0.01 in any of the sequenced populations compared to the parent ΔE3L virus. This variant, a point mutation encoding a Glu495Gly amino acid change in the viral DNA polymerase (E9L[Glu495Gly]), decreased in frequency from P10-P20, in contrast to the K3L[His47Arg] SNV (*Figure 1—figure supplement 2*). While the E9L[Glu495Gly] SNV reached high frequency at P10 (0.64), this variant alone did not provide a measurable increase in virus replication (*Figure 1—figure supplement 3*), suggesting that it may be non-adaptive, and could have accumulated as a hitchhiker mutation. These observations, and the lack of any other detectable genetic changes, suggest that virus gains in replication were dominated by

changes in K3L. Furthermore, the detection of only two high-frequency SNVs reflects the rarity of point mutations during poxvirus replication relative to RNA viruses and suggests that recombination-based mechanisms could be a major means of adaptation.

## ONT reads reveal precise K3L copy number and distributions of K3L$^{His47Arg}$ in individual genomes

To track the rise of the K3L$^{His47Arg}$ variant in virus populations containing K3L CNV, we needed a means to analyze large and repetitive arrays of sequence. Duplication of K3L produces breakpoints between flanking genetic regions that mark the boundaries of recombination (*Figure 2A*; *Elde et al., 2012*; *Cone et al., 2017*). We previously identified two distinct breakpoints flanking K3L in the P10 population that differ by only 3 bp (*Elde et al., 2012*). Each of these breakpoint pairs demarcates a duplicon approximately 500 bp in length (*Figure 2A*), and encompasses both the K3L open reading frame (ORF, 267 bp) and predicted K3L promoter (*Yang et al., 2010*). In heterogeneous virus populations, short (e.g. 150 bp) Illumina reads cannot discriminate the presence or absence of the K3L$^{His47Arg}$ variant either in single-copy K3L genomes or within multicopy arrays of K3L (*Figure 2A*). Therefore, we sequenced virus genomes using the ONT MinION sequencing platform and routinely generated reads with a mean length of ~3 kbp and a N50 between 5–8 kbp (*Table 1*). These reads reached a maximum aligned length of 40 kbp with a standard library preparation (see Materials and methods for further details) and allowed us to directly measure both K3L copy number and the presence or absence of K3L$^{His47Arg}$ in each K3L copy within individual virus genomes (*Figure 2A*).

Using ONT reads, we performed variant calling on the two high frequency SNVs (K3L$^{His47Arg}$ and E9L$^{Glu495Gly}$) from every fifth passage, yielding similar population-level allele frequencies to those estimated using Illumina MiSeq data for the same samples (*Figure 2B*, *Figure 2—figure supplement 1*; *Elde et al., 2012*). ONT sequencing error rates, which vary according to the specific k-mer being sequenced (*Jain et al., 2018*), are higher than in Illumina sequencing. To determine how error rates might influence variant allele frequency estimates, we calculated the proportions of sequencing errors (mismatches and deletions) at the 5-mers containing the wild-type or variant sequence for each SNV (K3L$^{WT}$: TA**T**GC; K3L$^{His47Arg}$: TA**C**GC; E9L$^{WT}$: AT**T**CG, E9L$^{Glu495Gly}$: AT**C**CG; see Materials and methods for further details). While error rates differ between ONT flow cell chemistries, we found that the highest error rate was only 2.6% (*Table 2*, *Figure 2—figure supplement 2*), supporting the utility of ONT reads for identifying high frequency SNVs in virus genomes. We also identified the same two K3L duplication breakpoints previously described in the P10 population (*Elde et al., 2012*) using nanopore reads from P5, P10, P15, and P20, which revealed that the ~500 bp duplicons were maintained in the virus population throughout passaging (*Table 3*). Given the high quality of the long ONT reads, we restricted subsequent data analysis to reads containing complete K3L arrays (reads that mapped ≥150 bp upstream and downstream of the duplicon), thereby excluding reads that only contain a subset of total K3L copies in a virus genome (see Materials and methods for further details).

The long read datasets revealed that K3L copy number expansions occurred as early as P5 (*Figure 2C*), consistent with Southern blot analysis (*Figure 1B*). Over the course of the next five passages, K3L copy number steadily increased, and nearly 70% of virus genomes contained multiple copies of the gene by P10 (*Figure 2C*). From passages 10 to 20, there was a modest shift toward higher copy number arrays, but the distribution of K3L copy number within the population appears to have reached a point near equilibrium. Throughout the experiment, over 90% of sequenced virus genomes contained between 1 and 5 copies of K3L (*Figure 2C*). These results, combined with the rapid collapse of copy number under relaxed selection (*Figure 1C*, *Elde et al., 2012*), are consistent with a fitness trade-off between additional K3L duplications and increased genome size at very high copy numbers. However, using ONT, we captured reads from genomes containing up to 16 total copies of K3L (*Figure 2—figure supplement 3*), confirming that rare, large gene arrays exist in the population. These virus genomes highlight the ability of long read sequencing to analyze large arrays of gene repeats.

High K3L copy number reads are likely underrepresented in our initial data sets, both because there is a higher probability of capturing a complete short array, and because an average sequencing read length of ~3 kbp limits the discovery of virus genomes with greater than 6–8 K3L copies. Therefore, we re-sequenced virus genomes from P15 using a DNA library preparation protocol

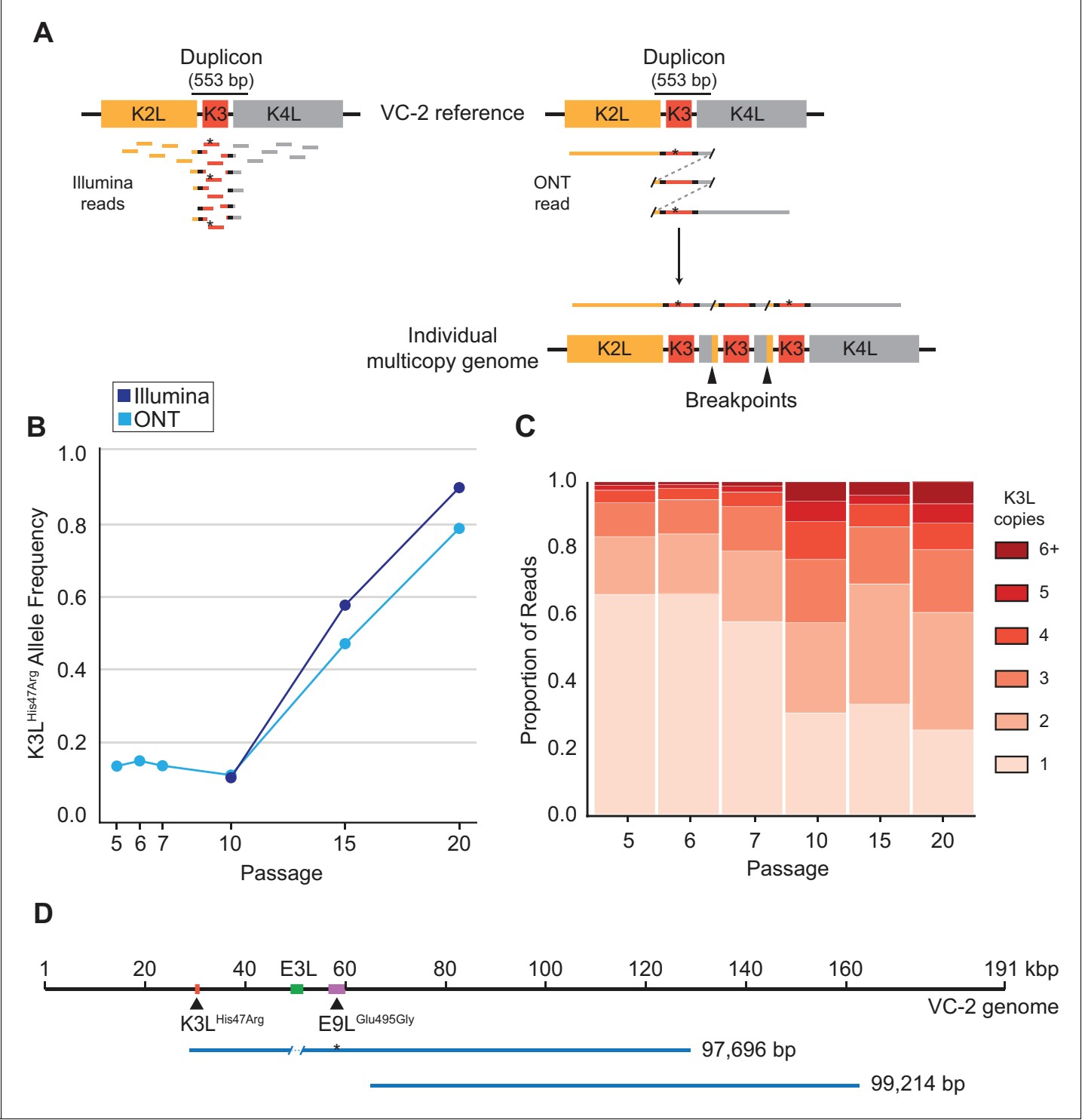

**Figure 2.** ONT reads capture SNVs and copy number expansions in individual virus genomes. (A) Representative structure of the K3L locus in the VC-2 reference genome is shown on top, with representative Illumina MiSeq and ONT MinION reads shown to scale below. The K3L$^{His47Arg}$ variant within reads is indicated by an asterisk. ONT reads that split and re-align to the K3L duplicon are indicative of individual multicopy arrays (shown below). Tandem duplication breakpoints flanking the duplicon are indicated by arrowheads. (B) Population-level K3L$^{His47Arg}$ allele frequency was estimated using Illumina or ONT reads from different passages. E9L$^{Glu495Gly}$ allele frequencies are shown in *Figure 2—figure supplement 1*. Error rate calculations for different flow cell chemistries are shown in *Figure 2—figure supplement 2*. (C) For each sequenced passage, K3L copy number was assessed within each ONT read that aligned at least once to the K3L duplicon (see Materials and methods for further details). Detailed plot of reads containing 6 + K3L copies is shown in *Figure 2—figure supplement 3*. (D) Representative reads from the specific long read preparation are depicted

*Figure 2 continued on next page*

*Figure 2 continued*

relative to the VC-2 reference genome. The locations of relevant genes are indicated by colored boxes (gene name above or below), and the locations of high frequency variants in K3L and E9L are indicated by arrowheads.

DOI: https://doi.org/10.7554/eLife.35453.014

The following source data and figure supplements are available for figure 2:

**Source data 1.** Single nucleotide variants in virus populations from Illumina or ONT datasets, used to generate *Figure 2B*.

DOI: https://doi.org/10.7554/eLife.35453.018

**Figure supplement 1.** E9L$^{Glu495Gly}$ variant dynamics.

DOI: https://doi.org/10.7554/eLife.35453.015

**Figure supplement 1—source data 1.** Data used to generate *Figure 2—figure supplement 1*.

DOI: https://doi.org/10.7554/eLife.35453.019

**Figure supplement 2.** Error rate profiles in ONT reads.

DOI: https://doi.org/10.7554/eLife.35453.016

**Figure supplement 3.** ONT reads capture high K3L copy number in vaccinia genomes.

DOI: https://doi.org/10.7554/eLife.35453.017

designed to generate extremely long reads (see Materials and methods for further details). The alternate protocol produced a mean read length of 9,392 bp (N50 = 19,288 bp), with a maximum aligned read length of 99,214 bp (*Figure 2D*). Even with increased read lengths, we did not recover larger proportions of high K3L copy number genomes in this dataset, suggesting that standard library preparations captured a representative sample of K3L copy number in virus populations. Using this specific long read preparation, we were also able to identify nearly 100 reads that span a ~ 30 kbp region separating the two high frequency single nucleotide variants in this population, K3L$^{His47Arg}$ and E9L$^{Glu495Gly}$ (*Figure 2D*). These extremely long sequencing reads enable the direct phasing of distant variants within single poxvirus genomes, and suggest that single reads may routinely capture entire poxvirus genomes in future studies.

## The K3L$^{His47Arg}$ variant rapidly homogenizes in multicopy vaccinia genomes

To determine how single nucleotide variants spread throughout tandem gene duplications within virus genomes, we assessed the presence or absence of the K3L$^{His47Arg}$ variant in each copy of K3L within our ONT reads (see Materials and methods for further details). Specifically, we categorized multicopy vaccinia genomes as containing either homogeneous K3L arrays, in which every K3L copy contains either the variant or wild-type sequence, or 'mixed' arrays, in which both K3L$^{WT}$ and K3L$^{His47Arg}$ copies are present in a single read. At P5, the K3L$^{His47Arg}$ SNV was observed almost exclusively in single-copy reads (four multicopy reads contained the SNV, compared to 304 single-copy reads; *Figure 3*), indicating that the variant likely originated in a single-copy genome. From P5 to P10, while the K3L$^{His47Arg}$ variant remained at a nearly constant population-level frequency (*Figure 2B*), we observed a slight increase in the proportion of multicopy genomes containing the SNV, which

**Table 1.** Summary of ONT sequencing datasets

| Population* | Total sequenced reads | Mean read length (bp) | Read length N50 (bp) | Total sequenced bases (Gbp) | Reads containing K3L |
|---|---|---|---|---|---|
| P5 | 239,737 | 2168 | 5932 | 0.52 | 1190 |
| P10 | 91,815 | 3523 | 7693 | 0.32 | 912 |
| P15 | 388,502 | 4493 | 6908 | 1.75 | 4317 |
| P20 | 94,050 | 2893 | 7702 | 0.27 | 789 |

*ONT sequencing datasets for all populations are available in *Table 1*-source data 1

DOI: https://doi.org/10.7554/eLife.35453.020

The following source data is available for Table 1:

**Table 1-Source data 1.** Complete summary of ONT sequencing datasets

DOI: https://doi.org/10.7554/eLife.35453.021

**Table 2.** Median sequencing error rates using various ONT flowcell chemistries

| Mutation and context (amino acid change) | R7.3 | R9 | R9.4 |
|---|---|---|---|
| TA[**T > C**]GC (His47Arg) | 0.023 | 0.023 | 0.005 |
| TA[**C > T**]GC (Arg47His) | 0.015 | 0.024 | 0.026 |
| AT[**T > C**]CG (Glu495Gly) | 0.014 | 0.018 | 0.009 |
| AT[**C > T**]CG (Gly495Glu) | 0.025 | 0.009 | 0.009 |

DOI: https://doi.org/10.7554/eLife.35453.022

was enriched for mixed, rather than homogeneous K3L$^{His47Arg}$ arrays (*Figure 3*). Although overall K3L copy number increased from P5 to P10, this pattern was driven mainly by an increase in homogeneous K3L$^{WT}$ arrays (*Figure 3*). Thus, even though the K3L$^{His47Arg}$ SNV was present in a high proportion of single-copy reads by P5, the scarcity of homogeneous K3L$^{His47Arg}$ arrays at P10 suggests that the SNV entered multicopy arrays through recombination, rather than gene amplification from single copies of K3L$^{His47Arg}$.

From passages 10–20, as the K3L$^{His47Arg}$ allele frequency drastically increased in the population (*Figure 2B*), we observed a very different pattern in the K3L arrays. By the last passages of the experiments, homogeneous K3L$^{His47Arg}$ arrays became increasingly prevalent, and were observed more frequently than mixed arrays (*Figure 3*). In contrast, when we simulated K3L$^{His47Arg}$ accumulation in passages 10–20 according to a binomial distribution, we observed a markedly lower prevalence of homogeneous K3L multicopy arrays (*Figure 3—figure supplement 1*, see Materials and methods for further details). One possible explanation for our observations is that amplification of K3L$^{His47Arg}$ or K3L$^{WT}$ copies drives the higher proportion of homogeneous arrays, and that sequencing errors at the K3L$^{His47Arg}$ site could lead us to label truly homogeneous multicopy K3L arrays as containing mixed alleles. Therefore, to address the impact of ONT sequencing errors on our observations of mixed arrays, we took two approaches. First, we sequenced the P15 population with a variety of flowcell chemistries. Despite having distinct error rate profiles, each chemistry yielded nearly identical distributions of homogeneous and mixed K3L arrays (*Figure 3—figure supplement 2*). Second, we simulated a population of K3L arrays that matched the copy number distribution of the P15 population, in which all arrays were homogeneous for K3L$^{WT}$ or K3L$^{His47Arg}$ alleles (see Materials and methods for further details). Then, as a proxy for sequencing errors, we randomly switched K3L$^{WT}$ alleles to K3L$^{His47Arg}$ (and vice versa) at a frequency equal to the median error rate for each flowcell chemistry used in our experiments. From this analysis, the experimental data consistently returned substantially higher fractions of mixed arrays compared to the 1000 simulations, confirming that our observed enrichment of mixed arrays was not an artifact of sequencing errors (*Figure 3—figure supplement 3*).

We further probed the mechanism underlying rapid homogenization of the K3L$^{His47Arg}$ variant by analyzing patterns of alleles within multicopy arrays from passages 10–20. In K3L arrays with 3 or 4 copies of the gene, we observed every possible combination of K3L$^{WT}$ and K3L$^{His47Arg}$ alleles at P15 (*Figure 3—figure supplement 4A–B*). A closer examination of 3-copy K3L arrays revealed steady homogenization of the K3L$^{His47Arg}$ SNV from P10 to P20 (*Figure 3—figure supplement 4A*). Mixed arrays remained prevalent in these populations, comprising between 23–42% of all 3-copy K3L arrays

**Table 3.** Structural variant breakpoint frequencies during passaging

| | | | Breakpoint frequency* | | | |
|---|---|---|---|---|---|---|
| Breakpoint | K2L break | K4L break | P5 | P10 | P15 | P20 |
| 1 | 30,284 | - | 0.76 | 0.69 | 0.76 | 0.66 |
| 1 | - | 30,837 | 0.76 | 0.63 | 0.72 | 0.62 |
| 2 | 30,287 | - | 0.14 | 0.06 | 0.10 | 0.08 |
| 2 | - | 30,840 | 0.12 | 0.04 | 0.09 | 0.05 |

*Due to sequencing errors, a proportion of reads do not match either breakpoint

DOI: https://doi.org/10.7554/eLife.35453.023

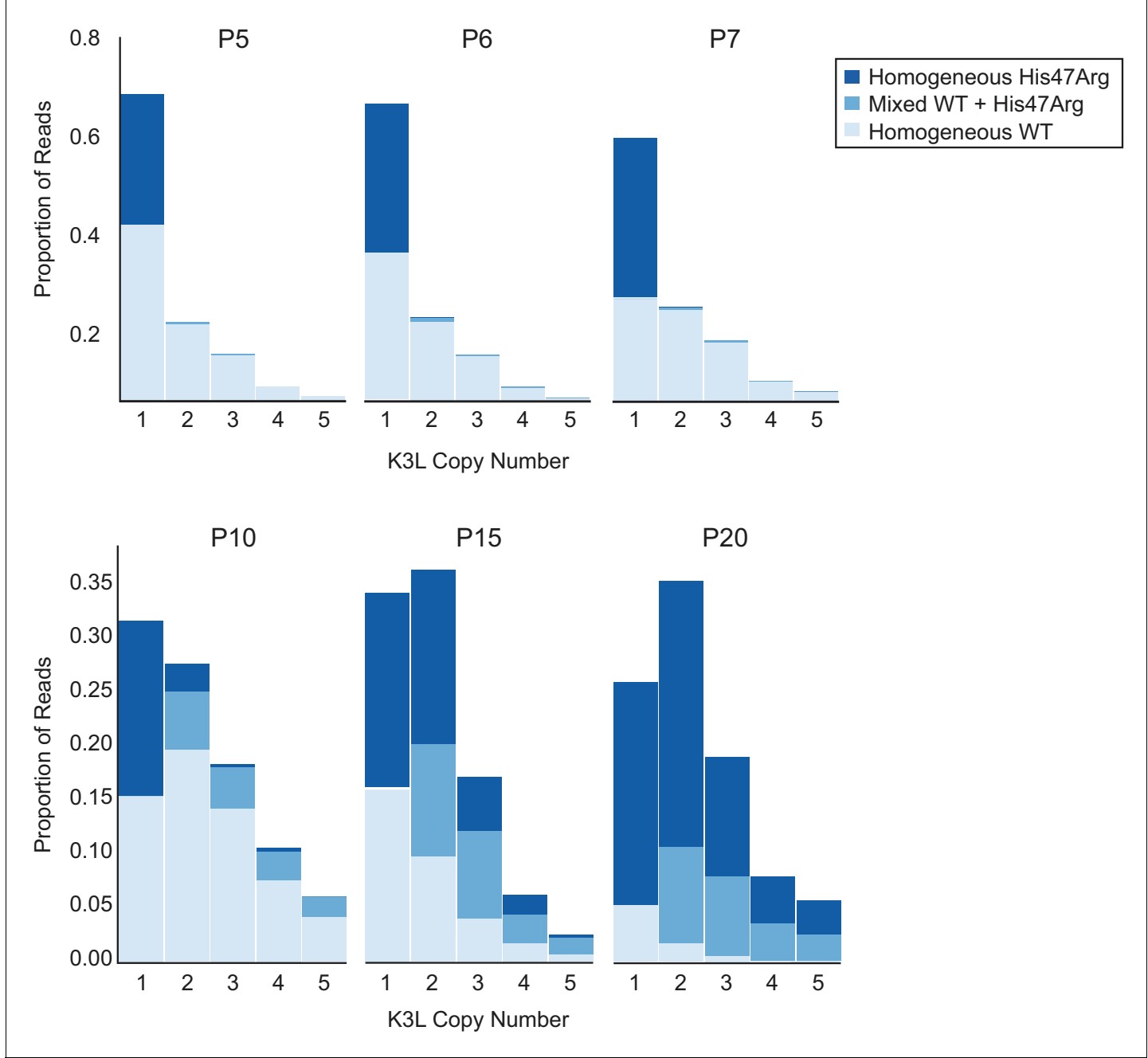

**Figure 3.** The K3L$^{His47Arg}$ variant homogenizes within multicopy arrays throughout experimental evolution. Stacked bar plots representing the proportions of mixed and homogeneous K3L arrays were generated from ONT reads for the indicated virus populations (passages are listed above each plot). The proportions of reads containing homogeneous K3L$^{WT}$, homogeneous K3L$^{His47Arg}$, or any combination of mixed alleles are shown for reads containing 1–5 K3L copies. A simulation of SNV accumulation under a binomial distribution is shown in *Figure 3—figure supplement 1*, and results from sequencing with different flow cell chemistries is shown in *Figure 3—figure supplement 2*. Simulations of the effects of ONT sequencing error rates on the identification of mixed and homogeneous arrays are shown in *Figure 3—figure supplement 3*, and the proportions of each combination of K3L alleles in 3, 4, and 5-copy arrays are shown in *Figure 3—figure supplement 4*.

DOI: https://doi.org/10.7554/eLife.35453.024

The following figure supplements are available for figure 3:

**Figure supplement 1.** Simulated accumulation of the K3L$^{His47Arg}$ SNV.

DOI: https://doi.org/10.7554/eLife.35453.025

**Figure supplement 2.** ONT flowcell chemistries do not affect observed proportions of homogeneous and mixed K3L arrays.

DOI: https://doi.org/10.7554/eLife.35453.026

*Figure 3 continued on next page*

*Figure 3 continued*

**Figure supplement 3.** ONT sequencing error rates do not affect observed proportions of homogeneous and mixed K3L arrays.
DOI: https://doi.org/10.7554/eLife.35453.027

**Figure supplement 4.** Multicopy K3L arrays contain diverse combinations of K3L$^{WT}$ and K3L$^{His47Arg}$ alleles.
DOI: https://doi.org/10.7554/eLife.35453.028

(*Figure 3*, *Figure 3—figure supplement 4A*). At P15, we also observed a similar overall proportion of various 3, 4, and 5-copy mixed arrays, suggesting that the trends observed for 3-copy arrays are as diverse, or potentially even more so, for mixed arrays at higher copy numbers (*Figure 3—figure supplement 4*). The abundance of mixed arrays, coupled with our observation of numerous allele combinations and the low point mutation rate of poxviruses (*Gago et al., 2009*; *Sanjuán et al., 2010*), strongly support a recombination-based mechanism of variant homogenization. Consistent with this idea, negligible sequencing error rates observed in our identification of K3L alleles demonstrate that mixed arrays are not technical artifacts (*Figure 2—figure supplement 2*, *Figure 3—figure supplement 3*) and that replacement of K3L$^{WT}$ arrays with pure K3L$^{His47Arg}$ arrays is not the primary means of variant homogenization (*Figure 3*). These findings reveal recombination-driven genetic homogenization in the rapid rise of adaptive variation in homologous sequences.

Throughout the passaging experiment, the K3L$^{His47Arg}$ SNV was nearly equally distributed throughout K3L arrays, regardless of K3L copy number (*Figure 4A*). To determine whether the presence of homogeneous K3L arrays, either K3L$^{WT}$ or K3L$^{His47Arg}$, influenced this pattern, we reanalyzed P15 virus genomes after removing all homogeneous K3L arrays from the dataset (*Figure 4B*). We observed the same striking pattern of homogeneity for the SNV, regardless of array copy number or a copy's position within the array. Because the same K3L$^{His47Arg}$ SNV frequency is observed even when homogeneous K3L$^{WT}$ or K3L$^{His47Arg}$ genomes are excluded, these results suggest that once the K3L$^{His47Arg}$ variant entered multicopy genomes, variant accumulation was independent of copy number.

## Recombination and selection drive patterns of K3L$^{His47Arg}$ homogenization

To investigate the influence of intergenomic recombination between co-infecting viruses on gene homogenization, we conducted serial infections at various multiplicity of infection (MOI). We repeated passages 11 to 15 using a range of MOI from 1.0 to 0.001 (the original experiments are MOI = 0.1), in order to determine whether increasing or virtually eliminating the occurrence of intergenomic recombination would affect the accumulation of the K3L$^{His47Arg}$ variant. While intergenomic recombination has been observed following passaging of wild-type viruses at a low MOI (0.02; *Qin and Evans, 2014*), our lowest MOI (0.001), combined with the reduced replication ability of these virus populations (*Figure 1A*), greatly diminishes the probability of co-infection even over the course of a 48 hour passage. Analysis of all four P15 populations returned similar distributions of K3L copy number, as well as distributions of homogeneous and mixed arrays, regardless of MOI (*Figure 5*). Thus, even when co-infection is rare at the lowest MOI, these distributions are consistent, suggesting that patterns of variant accumulation are robust to changes in the probability of intergenomic recombination. Therefore, intragenomic recombination during replication from single virus infections is likely the main source for rapid homogenization of the K3L$^{His47Arg}$ variant within gene arrays. Frequent crossover recombination events between virus genomes could contribute to the diversity of mixed K3L arrays. However, the initial presence of multicopy arrays containing mixed alleles, quickly followed by the predominance of multicopy arrays homogeneous for the variant (*Figure 3*) is also consistent with abundant intragenomic recombination resulting in a process reminiscent of gene conversion.

Gene conversion is a driving force behind sustained sequence identity among repeated sequences, promoting concerted evolution (*Chen et al., 2007*; *Ohta, 2010*). While the precise role and outcomes of recombination during the spread of the K3L$^{His47Arg}$ variant are difficult to test without a clear understanding of the recombination machinery in poxviruses (*Gammon and Evans, 2009*), two lines of evidence highlight the importance of natural selection for homogenization of the K3L$^{His47Arg}$ variant in multicopy gene arrays. First, we repeated passages 11 to 15 under relaxed selection on K3L by infecting BHK cells, in which neither the K3L$^{His47Arg}$ variant nor K3L CNV provides a

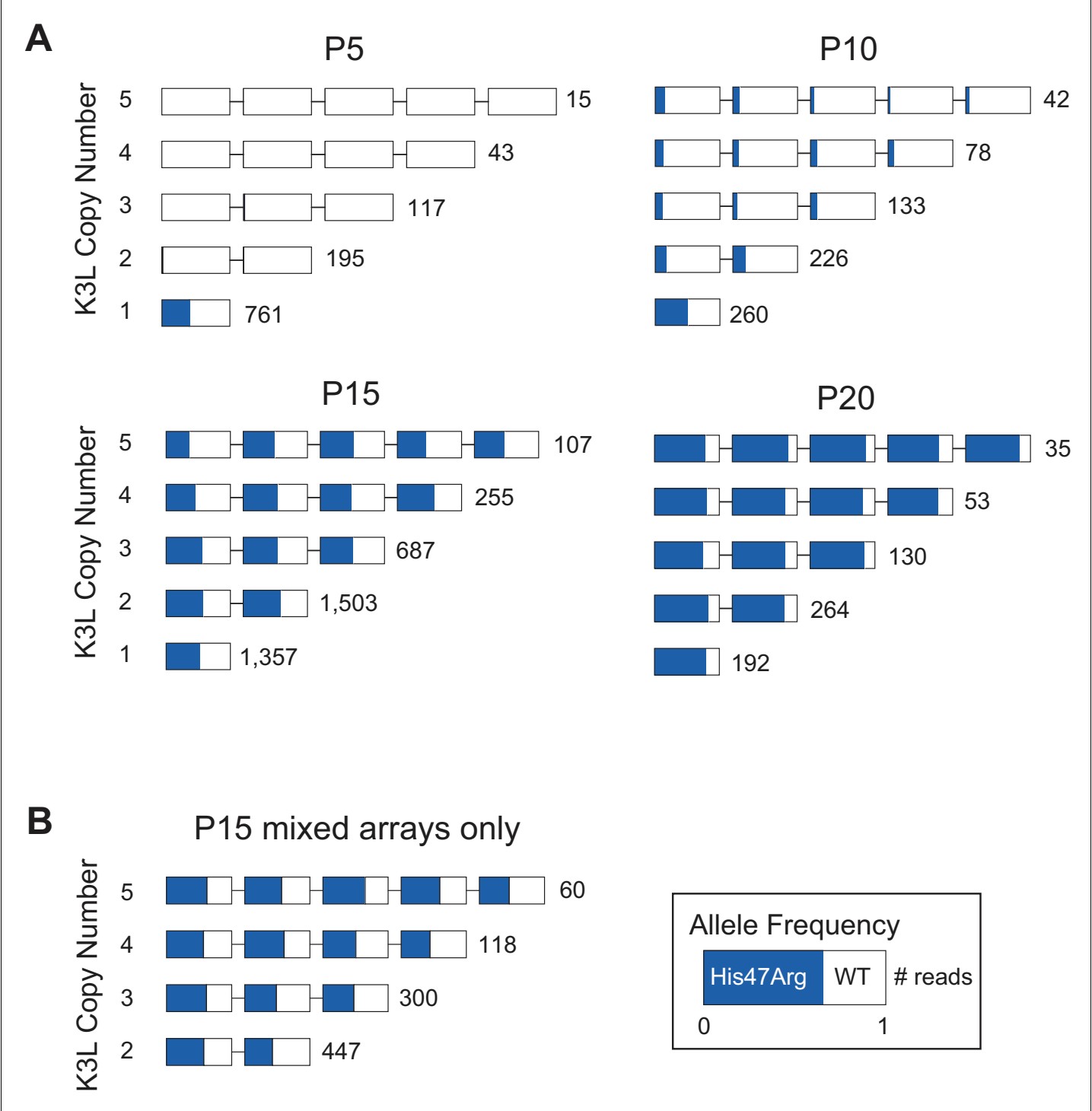

**Figure 4.** The K3L[His47Arg] variant homogenizes in K3L arrays regardless of copy number. (**A**) ONT reads from every 5th passage were grouped by K3L copy number, and each K3L copy was assessed for the presence or absence of the K3L[His47Arg] SNV. Reads containing 1–5 K3L copies are shown. (**B**) Using reads from the P15 population, homogeneous K3L arrays were removed from the dataset, and K3L[His47Arg] SNV frequency was plotted exclusively in mixed arrays. The number of reads of each copy number is indicated to the right of each row. Reads are oriented 5' to 3' relative to the VC-2 reference sequence, and the K3L[His47Arg] allele frequency in each copy is indicated in blue.

DOI: https://doi.org/10.7554/eLife.35453.029

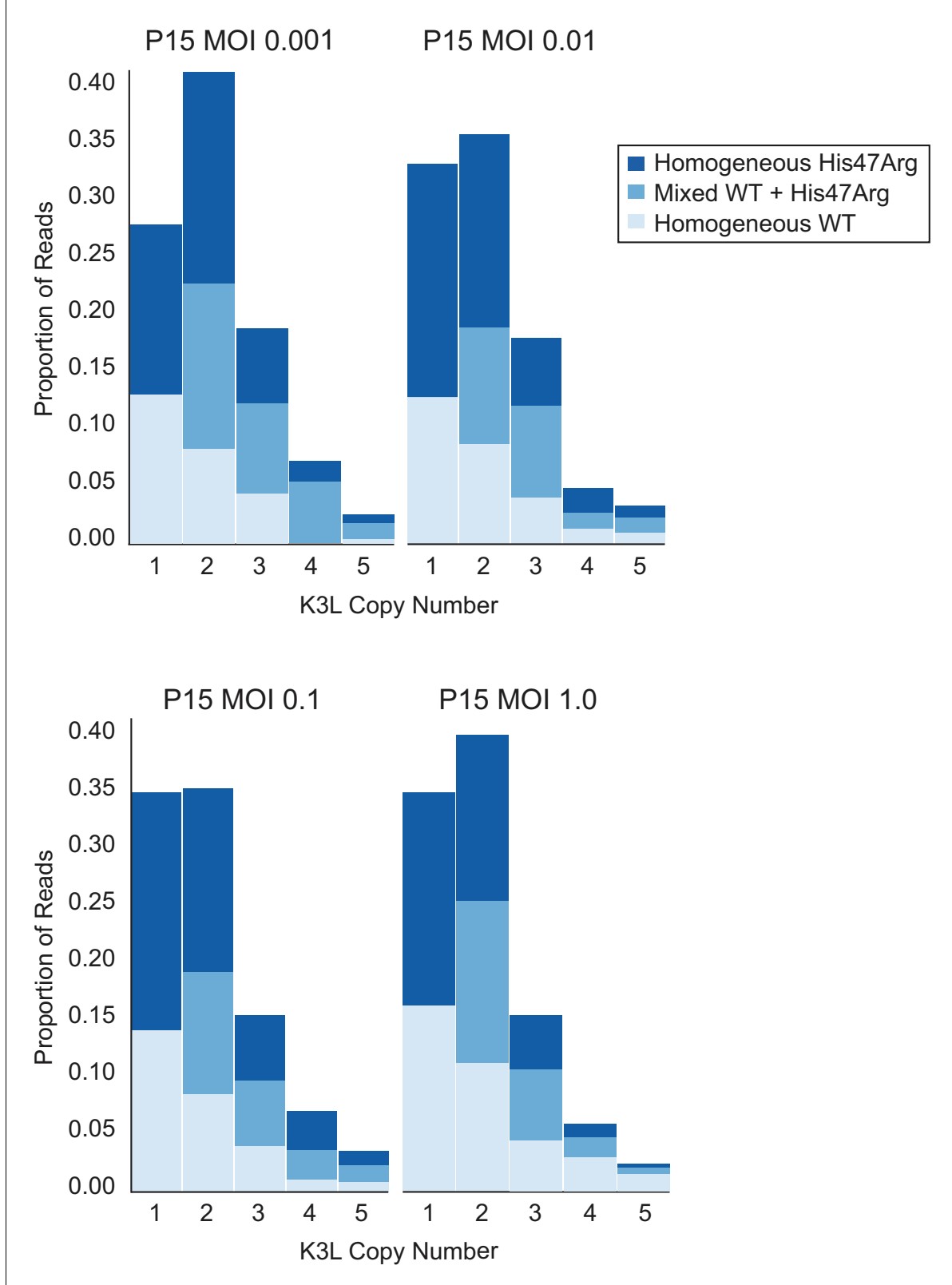

**Figure 5.** K3L[His47Arg] homogenization within multicopy arrays is independent of intergenomic recombination rate. The P10 population was serially passaged in HeLa cells at different MOIs (listed above each plot), and each of the resulting P15 populations was sequenced with ONT. Stacked bar plots representing the proportions of mixed and homogeneous arrays were generated as in *Figure 3*.
DOI: https://doi.org/10.7554/eLife.35453.030

measurable replication benefit (*Figure 1—figure supplement 1*; *Elde et al., 2012*). Consistent with related protocols of plaque purification in BHK cells (*Figure 1C*), we observed a uniform reduction of K3L copy number in the population following five passages in BHK cells (P15-BHK; *Figure 6A*). Further analysis of virus genomes from the P15-BHK population revealed that the K3L$^{His47Arg}$ variant had not increased in frequency compared to P10, in contrast to its accumulation in HeLa cells (*Figure 6B*). Additionally, the proportion of reads homogeneous for K3L$^{His47Arg}$ decreased following passaging in BHK cells (*Figure 6C*), as opposed to the rapid homogenization observed in HeLa cells. These results suggest that variant accumulation and homogenization are dependent on selective pressure imposed by the host environment.

Next, we considered how selection influenced the accumulation of the K3L$^{His47Arg}$ SNV relative to variation in the population at two other loci. Unlike the K3L$^{His47Arg}$ variant, the only other observed point mutation, E9L$^{Glu495Gly}$, did not provide a replication benefit in human cells, indicating differential selection for each SNV (*Figure 1D*, *Figure 1—figure supplement 3*). Indeed, only the K3L$^{His47Arg}$ variant reached near-fixation following passaging in human cells, while the E9L$^{Glu495Gly}$ variant instead decreased over the course of passaging in both HeLa and BHK cells (*Figure 2—figure supplement 1* and *Figure 6B*). Similarly, the presence of two distinct recombination breakpoints (located three base pairs apart) allowed us to compare structural variation expected to be neutral relative to amino acid position 47 in K3L (*Figure 2A*), because both breakpoints fall outside of the K3L ORF and predicted promoter sequence (*Yang et al., 2010*). Across all analyzed populations, we observed that one breakpoint was dominant over the other, and that the frequency of these breakpoints did not appreciably change over the course of passaging in either cell line (*Table 3* and *Figure 6D*). The consistency of these neutral genetic changes sharply contrasts the rapid accumulation of the K3L$^{His47Arg}$ variant in response to selection, providing further support for the idea that selection is required to drive rapid homogenization of the K3L$^{His47Arg}$ allele. Additionally, we observed that the frequencies of the K3L$^{His47Arg}$ SNV and the K3L breakpoints are uncoupled (*Figure 6B*, *Figure 6D*), despite their proximity in the genome (*Figure 2A*). This observation supports gene conversion as a potentially major mechanism driving only the SNV to high frequency, rather than reciprocal recombination linking a particular breakpoint to the point mutation as it accumulated within the population. Taken together, our results support a model of adaptation resembling gene conversion, in which a beneficial variant that enters an amplified gene array can be rapidly spread among the remaining copies through recombination and selection (*Figure 7*).

## Discussion

In this study, we investigated how a virus population evolves given the simultaneous presence of distinct adaptive variants at a single locus. In vaccinia virus populations harboring recombination-driven gene copy number amplifications and a beneficial point mutation in the same gene, courses of experimental evolution revealed a process of variant homogenization resembling gene conversion. Repeated gene amplification likely also contributes to the accumulation of the beneficial SNV; however, the large proportions and various patterns of mixed allele combinations in multicopy arrays suggest that this is not the only mechanism of genome diversification. Furthermore, while we cannot directly distinguish between gene conversion and crossover events, the consistency of neutral variants in close proximity to the rapidly homogenized SNV strongly support a model of recombination-driven homogenization. Future analyses might inform additional means of adaptation, but these results support gene conversion as a potentially critical mechanism underlying rapid homogenization of a SNV within repeated gene copies. This process could be a unique adaptive feature of large DNA viruses, because evolution through mechanisms of gene duplication are widespread in DNA viruses (*McLysaght et al., 2003*; *Shackelton and Holmes, 2004*; *Filée, 2009*; *Elde et al., 2012*; *Filée, 2015*; *Gao et al., 2017*), but rare in RNA viruses (*Simon-Loriere and Holmes, 2013*). For DNA viruses, which possess significantly lower point mutation rates than RNA viruses (*Gago et al., 2009*; *Sanjuán et al., 2010*), the rapid fixation of rare beneficial variants within multiple gene copies could be key to the process of adaptation.

A major outcome from genetic homogenization of beneficial point mutations in multicopy genes might be an enhanced persistence of large gene families. Under this model, the rapid spread of point mutations in gene arrays would counter the advantages of single or low-copy genomes enriched for the SNV to dominate DNA virus populations. Among poxviruses, for example, nearly

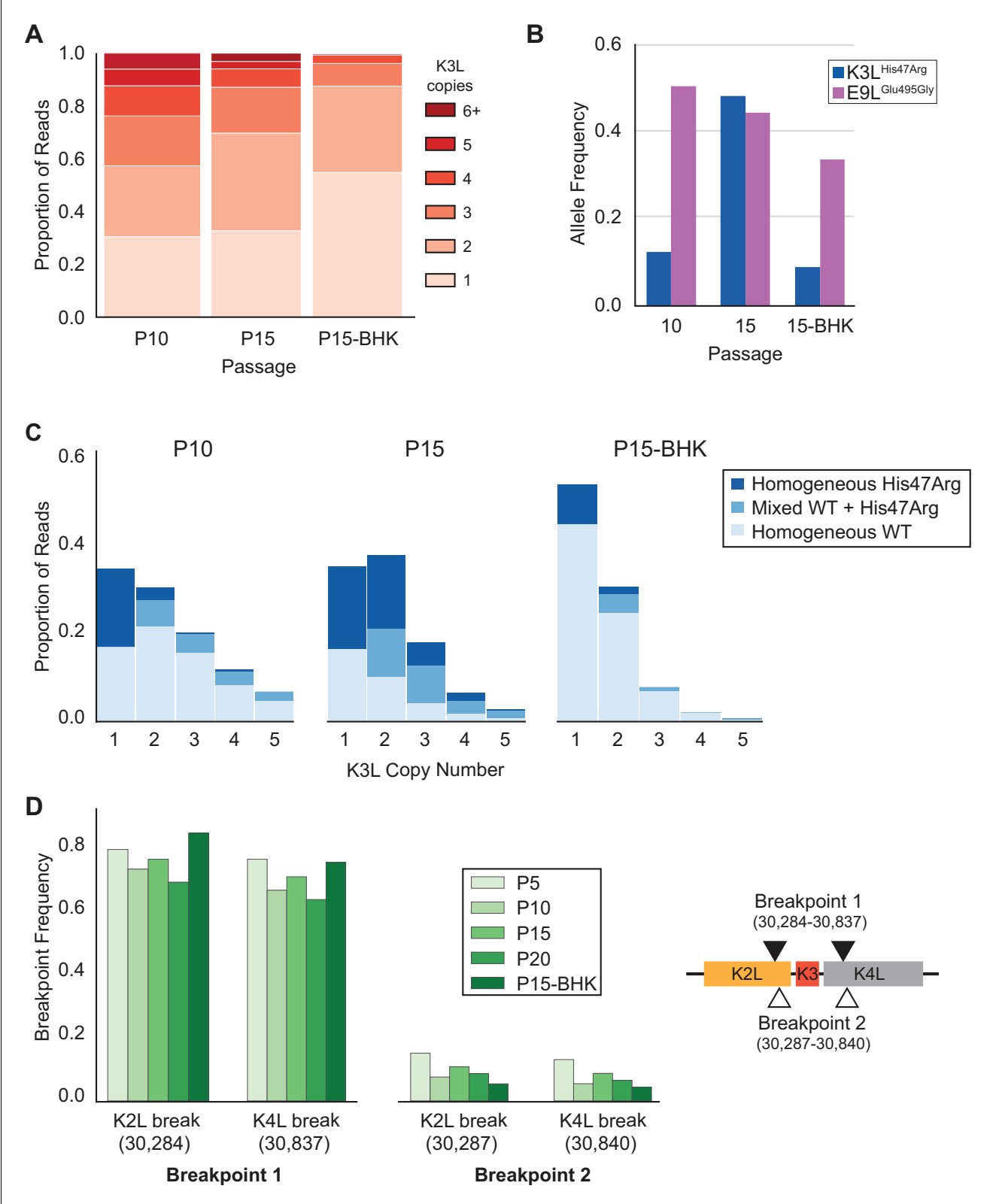

**Figure 6.** K3L^His47Arg variant homogenization is dependent on selection. The P10 population was serially passaged five times in BHK cells (MOI = 0.1, 48 hr; P15-BHK). P10 and P15 data are included from previous figures for comparison with P15-BHK. (**A**) K3L copy number was assessed for all sequenced reads that unambiguously aligned to K3L at least once, as in **Figure 2C**. (**B**) K3L^His47Arg and E9L^Glu495Gly allele frequencies in each population were estimated using ONT reads, as in **Figure 2B**. Allele frequencies for all sequenced populations are included in **Figure 6—source data**

*Figure 6 continued on next page*

*Figure 6 continued*

*1*. (**C**) Stacked bar plots representing the proportions of mixed and homogeneous arrays were generated from sequenced ONT reads, as in *Figure 3*. (**D**) ONT reads were assessed for the presence of each breakpoint (shown relative to the genome to the right) by aligning reads to a query sequence containing K3L using BLAST and extracting the starts and ends of individual alignments to the K3L duplicon. Due to sequencing errors, a proportion of reads do not match either breakpoint 1 or breakpoint 2.

DOI: https://doi.org/10.7554/eLife.35453.031

The following source data is available for figure 6:

**Source data 2.** Data used to generate *Figure 6D*.
DOI: https://doi.org/10.7554/eLife.35453.032
**Source data 1.** Single nucleotide variants in all sequenced virus populations from Illumina or ONT datasets.
DOI: https://doi.org/10.7554/eLife.35453.033

half of the Canarypox genome consists of 14 gene families, which may have been regularly shaped by mechanisms of genetic homogenization (*Afonso et al., 2000*; *Tulman et al., 2004*). Within the emerging classes of giant viruses, the Bodo saltan virus is notable for a large gene family of 148 ankyrin repeat proteins at the ends of its linear genome, with some copies being nearly identical (*Deeg et al., 2018*). In cases like these, point mutations that are sampled among tandem arrays of genes might quickly spread to fixation through homogenization. Indeed, repeated homogenization through gene conversion has been suggested as the process driving concerted evolution of viral genes (*Hughes, 2004*), which has been observed in nanoviruses (*Hughes, 2004*; *Hu et al., 2007*; *Savory and Ramakrishnan, 2014*), baculoviruses (*de Andrade Zanotto and Krakauer, 2008*), and Epstein-Barr virus, a human herpesvirus (*Ba Abdullah et al., 2017*). In these studies, comparing naturally existing strains led the authors to propose cases of concerted evolution, however, short read sequencing of fixed populations restricted the possibility of investigating underlying mechanisms. In contrast, our experimental system allowed us to track an actively evolving virus population, and using long DNA sequencing reads, uncover a model consistent with gene conversion driving the rapid homogenization of a variant within gene arrays. We present a set of tools to study poxviruses and other DNA virus populations to determine whether similar mechanisms of diversification underlie adaptation during natural virus infections.

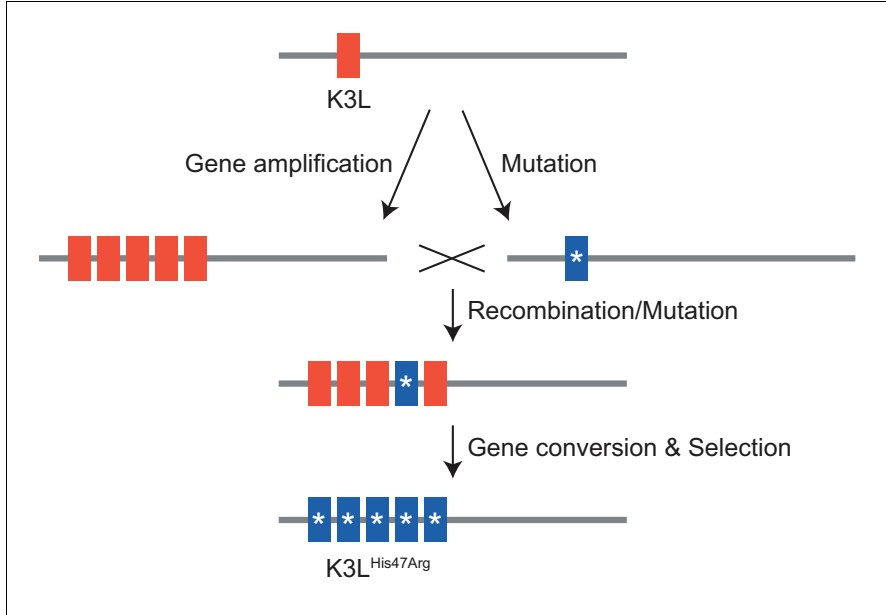

**Figure 7.** Model of K3L$^{His47Arg}$ homogenization within K3L CNV via gene conversion.
DOI: https://doi.org/10.7554/eLife.35453.034

Our work also demonstrates the power of long read sequencing to perform high resolution analyses of complex genome dynamics. Using the Oxford Nanopore Technologies platform, we investigated two simultaneous adaptations at single-genome resolution. This type of analysis provides a framework to definitively determine the sequence content of tandem gene duplications, and accurately call variants within these repetitive regions. Additionally, we demonstrate the ability to phase extremely distant genetic variants, and the longest reads we obtained suggest that entire poxvirus genomes could routinely be captured in single reads. Sequencing entire DNA virus genomes with this level of detail could expand our understanding of DNA virus adaptation as a population evolves, either in an experimental system or during infection of a host. Together, these methods allow for high-resolution analyses of complex genomes and could be used to explore the evolution of diverse organisms in new and exciting detail.

# Materials and methods

**Key resources table**

| Reagent type (species) or resource | Designation | Source or reference | Identifiers | Additional information |
|---|---|---|---|---|
| Gene (*Vaccinia virus*) | K3L | NA | NCBI_Gene ID:3707649 | |
| Strain, strain background (*Vaccinia virus*) | VC-2, Copenhagen | (*Goebel et al., 1990*) PMID: 2219722 | NCBI_txid:10249; NCBI_GenBank:M35027.1 | |
| Strain, strain background (*Vaccinia virus*) | ΔE3L, Copenhagen | (*Beattie et al., 1995*) PMID: 7527085 | | |
| Cell line (*Homo sapiens*) | HeLa | Other | | Obtained from Geballe lab, University of Washington |
| Cell line (*Mesocricetus auratus*) | BHK | Other | | Obtained from Geballe lab, University of Washington |
| Commercial assay or kit | Covaris g-TUBE | Covaris, Inc. | Catalog no: 520079 | |
| Commercial assay or kit | DIG High-Prime DNA Labeling and Detection Starter Kit II | Roche | Catalog no: 11585614910 | |
| Commercial assay or kit | Nextera XT DNA library preparation kit | Illumina | Catalog no: FC-131–1024 | |
| Commercial assay or kit | SQK-NSK007; SQK-LSK208; SQK-LSK308; SQK-RAD002 | Oxford Nanopore Technologies | Catalog no: SQK-NSK007; SQK-LSK208; SQK-LSK308; SQK-RAD002 | |
| Commercial assay or kit | FLO-MIN104; FLO-MIN106; FLO-MIN107 | Oxford Nanopore Technologies | Catalog no: FLO-MIN104; FLO-MIN106; FLO-MIN107 | |
| Chemical compound, drug | DMEM | HyClone, VWR | Catalog no: 16777–129 | |
| Chemical compound, drug | FBS | HyClone, VWR | Catalog no: 26-140-079 | |
| Chemical compound, drug | Penicillin-streptomycin | GE Life Sciences, VWR | Catalog no: 16777–164 | |
| Chemical compound, drug | SG-2000 | GE Life Sciences, VWR | Catalog no: 82024–258 | |
| Software, algorithm | GraphPad Prism | GraphPad Software | | |
| Software, algorithm | BWA-MEM | (*Li, 2013*) | v0.7.15 | arxiv.org/abs/1303.3997 |
| Software, algorithm | *samblaster* | (*Faust and Hall, 2014*) PMID: 24812344 | v0.1.24 | https://github.com/GregoryFaust/samblaster |
| Software, algorithm | *freebayes* | (*Garrison and Marth, 2012*) | v1.0.2–14 | arxiv.org/abs/1207.3907 |
| Software, algorithm | Metrichor | Oxford Nanopore Technologies | v2.40 | |

*Continued on next page*

*Continued*

| Reagent type (species) or resource | Designation | Source or reference | Identifiers | Additional information |
|---|---|---|---|---|
| Software, algorithm | Albacore | Oxford Nanopore Technologies | v1.2.4 | |
| Software, algorithm | *poretools* | (*Loman and Quinlan, 2014*) PMID: 25143291 | v0.6.0 | https://github.com/arq5x/poretools |
| Software, algorithm | Porechop | Other | v0.2.3 | https://github.com/rrwick/Porechop |
| Software, algorithm | *nanopolish* | (*Loman et al., 2015*) PMID: 26076426 | v0.8.4 | https://github.com/jts/nanopolish |
| Software, algorithm | source code | this paper | | See Materials and methods, https://github.com/tomsasani/vacv-ont-manuscript; copy archived at https://github.com/elifesciences-publications/vacv-ont-manuscript) |
| Software, algorithm | raw sequencing data | this paper | SRP128569; SRP128573; DOI: 10.5281/zenodo.1319732 | See Materials and methods |
| Software, algorithm | raw sequencing data | (*Elde et al., 2012*) PMID: 22901812 | SRP013146 | |

## Cells

HeLa and BHK cells were maintained in Dulbecco's modified Eagle's medium (DMEM; HyClone, Logan, UT) supplemented with 10% fetal bovine serum (HyClone), 1% penicillin-streptomycin (GE Life Sciences, Chicago, IL), and 1% stable L-glutamine (GE Life Sciences). Cell lines were authenticated by STR analysis of 24 loci (DNA Sequencing Core Facility, University of Utah, Salt Lake City, UT) for HeLa cells, and species-specific PCR for both HeLa and BHK cells as previously described (*Steube et al., 2008*). Both cell lines tested negative for mycoplasma contamination, using the MycoSensor PCR Assay Kit (Agilent Technologies, Inc., Santa Clara, CA) according to the manufacturer's protocol.

## Experimental evolution

The P5-P10 populations of vaccinia virus were previously established following serial passages of the ΔE3L virus (*Beattie et al., 1995*) in HeLa cells (*Elde et al., 2012*). Briefly, 150 mm dishes were seeded with an aliquot from the same stock of HeLa cells ($5 \times 10^6$ cells/dish) and infected (MOI = 1.0 for P1, and MOI = 0.1 for subsequent passages) for 48 hr. Cells were then collected, washed, pelleted, and resuspended in 1 mL of media. Virus was released by one freeze/thaw cycle followed by sonication. P10 (replicate C passage 10 in *Elde et al., 2012*) virus was expanded in BHK cells, and titer determined by 48 hr plaque assay in BHK cells. Passages 11–20 were performed as above, starting with the P10 virus population. Following passage 20, replication ability was assayed simultaneously by 48 hr infection (MOI = 0.1) in at least triplicate in HeLa cells.

For the intergenomic recombination passages, P10 virus was passaged in HeLa cells five times as above at a range of MOI (MOI = 0.001–1.0 as indicated) for 48 hr. For BHK passages, the P10 virus population was passaged five times as above, using an aliquot from the same stock of BHK cells ($5 \times 10^6$ cells/dish) infected (MOI = 0.1) for 48 hr.

All replication comparisons were performed on biological triplicates, assessing virus titers by 48 hr plaque assay in BHK cells performed with at least three technical replicates, as shown in the source data associated with each figure. Statistical analyses were performed using GraphPad Prism (GraphPad Software, La Jolla, CA).

## Southern blot analysis

Viral DNA from purified viral cores was digested with EcoRV (New England Biolabs, Ipswich, MA), and separated by agarose gel electrophoresis. DNA was transferred to nylon membranes (GE Life Sciences) using a vacuum transfer, followed by UV-crosslinking. Blots were probed with PCR-

amplified K3L using the DIG High-Prime DNA Labeling and Detection Starter Kit II (Roche, Basel, Switzerland) according to the manufacturer's protocol.

## Isolation and testing of plaque purified clones

BHK cells were infected for 48 hr with dilutions of P15 or P20 virus, and overlayed with 0.4% agarose. Single plaques were harvested and transferred to new BHK dishes, and resulting wells harvested after 48 hr. Virus was released by one freeze/thaw cycle followed by sonication. The process was repeated three additional times for a total of four plaque purifications. Viral DNA was extracted from individual clones from each population as previously described (*Esposito et al., 1981*) from infected BHK cells (MOI = 0.1) 24 hr post-infection, and assessed for K3L CNV and the K3L$^{His47Arg}$ SNV by PCR and Sanger sequencing.

One clone of each genotype (single-copy K3L$^{WT}$ or K3L$^{His47Arg}$, and multicopy K3L$^{WT}$ or K3L$^{His47Arg}$) was expanded in BHK cells. Replication ability was assessed by 48 hr infection (MOI = 0.1) in triplicate in either HeLa or BHK cells. Virus titers were determined by 48 hr plaque assay in BHK cells performed with at least three technical replicates.

## Deep sequencing of viral genomes

### Illumina

Total viral genomic DNA was collected as above (*Esposito et al., 1981*). Libraries were constructed using the Nextera XT DNA sample prep kit (Illumina, Inc., San Diego, CA). Barcoded libraries were pooled and sequenced on an Illumina MiSeq instrument (High-Throughput Genomics Core Facility, University of Utah). Reads were mapped to the Copenhagen reference strain of vaccinia virus (VC-2; accession M35027.1, modified on poxvirus.org; *Goebel et al., 1990*) using default BWA-MEM v0.7.15 (*Li, 2013*) parameters. PCR duplicates were removed using *samblaster* (*Faust and Hall, 2014*). Variant calling was performed using *freebayes* v1.0.2–14 (*Garrison and Marth, 2012*), using the following parameters:

```
freebayes -f $REF $BAM –pooled-continuous -C 1 –genotype-qualities –report-geno-
type-likelihood-max -F 0.01.
```

### Oxford Nanopore Technologies

Virus particles were isolated from infected BHK cells (MOI = 1.0) 24 hr post-infection, and viral cores were purified by ultracentrifugation through a 36% sucrose cushion at 60,000 rcf for 80 min. Total viral genomic DNA was extracted from purified cores as above (*Esposito et al., 1981*). Purified DNA was then size-selected in a Covaris g-TUBE (Covaris, Inc., Woburn, MA) 2 × 1 min at 6000 rpm. Sequencing libraries for the P10, P15, and P20 populations were prepared using the ONT SQK-NSK007 kit and sequenced on R9 chemistry MinION flow cells (FLO-MIN104); libraries for the P5, P6, and P7 populations were prepared using the ONT SQK-LSK208 kit and sequenced on R9.4 chemistry flow cells (FLO-MIN106); libraries for the P15 MOI 0.01, P15 MOI 0.1, and P15-BHK populations were prepared using the SQK-LSK308 kit and sequenced on R9.5 chemistry flow cells (FLO-MIN107); libraries for the P15 MOI 0.001 and P15 MOI 1.0 populations were sequenced using both R9.4 and R9.5 chemistry flow cells; additional libraries for the P15 population were sequenced using R9.5 chemistry flow cells (Oxford Nanopore Technologies Ltd., Oxford, UK). For the specific long read library preparation, we used purified, un-sheared P15 viral DNA and a SQK-RAD002 sequencing kit; libraries were sequenced on a FLO-MIN106 flow cell. All sequencing reactions were performed using a MinION Mk1B device and run for 48 hr; base calling for R9 reactions was performed using the Metrichor cloud suite (v2.40), while a command line implementation of the Albacore base caller (v1.2.4) was used to base call data from the remaining sequencing runs. For Albacore base calling on R9.4 runs, the following command was used:

```
read_fast5_basecaller.py -k SQK-LSK208 -f FLO-MIN106 -o fast5 -t 16 r -i
$RAW_FAST5_DIRECTORY.
```

For Albacore base calling on R9.5 runs, the following command was used:

```
full_1dsq_basecaller.py -k SQK-LSK308 -f FLO-MIN107 -o fastq,fast5 -t 16 r -i
$RAW_FAST5_DIRECTORY.
```

For all R9 and R9.4 data, FASTQ sequences were extracted from base-called FAST5 files using *poretools* v0.6.0 (*Loman and Quinlan, 2014*), while FASTQ were automatically generated by Albacore during base-calling on the R9.5 populations. Prior to alignment, adapter trimming on all ONT reads was performed using Porechop (https://github.com/rrwick/Porechop). Only the highest quality reads (2D and $1D^2$ for R9/R9.4 and R9.5 chemistries, respectively) were used for downstream analysis. Pooled FASTQ files for each sample were then aligned to the VC-2 reference genome with BWA-MEM v0.7.15 (*Li, 2013*), using the default settings provided by the -x ont2d flag. Population-level estimates of SNV frequencies were determined from our nanopore data using *nanopolish* v0.8.4 (*Loman et al., 2015*).

## Copy number and allele frequency estimation

Custom Python scripts (www.github.com/tomsasani/vacv-ont-manuscript [copy archived at https://github.com/elifesciences-publications/vacv-ont-manuscript] DOI: 10.5281/zenodo.1320424) were used to calculate both K3L copy number and K3L$^{His47Arg}$ allele frequency within individual aligned virus reads. Briefly, to identify individual ONT reads containing K3L, we first selected all reads that aligned at least once to the duplicon containing K3L. We next categorized ONT reads containing K3L as single-copy or multicopy. Reads that unambiguously aligned once to the K3L locus were classified as single-copy. The VC-2 reference contains a single K3L gene; therefore, if more than one distinct portion of a read aligned to the full length of K3L, that read was instead classified as multicopy. For every one of a read's alignments to K3L, we examined the read base aligned to reference position 30,490. If the read base matched the reference, we catalogued that alignment as being K3L$^{WT}$, and if the read base was a cytosine, we catalogued it as K3L$^{His47Arg}$. We filtered reads to include only those that aligned to the K3L duplicon, as well as 150 bp of unique VC-2 sequence upstream and downstream of the K3L duplicon, to ensure that the read fully contained the estimated number of K3L copies and was derived from a single vaccinia genome. Finally, we removed sequencing reads that contained one or more truncated alignments to the K3L duplicon, one or more alignments to K3L with mapping qualities less than 20, or one or more alignments to K3L with a non-reference or non-K3L$^{His47Arg}$ base (i.e., not a C or T) at reference position 30,490.

## Breakpoint characterization

To characterize the proportions of K3L duplicon breakpoint pairs in the ONT data, we first extracted all reads from each population that unambiguously aligned to K3L at least once, and that also aligned to unique sequence 150 base pairs up- and downstream of the K3L duplicon. We then aligned a 1000 bp query sequence (containing VC-2 reference sequence from genomic position 30,000 to 31,000) to each of the extracted reads using BLAST (*Zhang et al., 2000*). For every read, we extracted the start and end coordinates of all high-quality alignments to the K3L query, ±50 bp from the expected K3L breakpoints (*Elde et al., 2012*).

## Estimating ONT sequencing error

We analyzed ONT sequencing accuracy at each k-mer centered on the reference or non-reference base for the K3L$^{His47Arg}$ or E9L$^{Glu495Gly}$ SNVs in our ONT datasets (K3L$^{WT}$: TA**T**GC, K3L$^{His47Arg}$: TA**C**GC, E9L$^{WT}$: AT**T**CG, E9L$^{Glu495Gly}$: AT**C**CG). For each instance of the 5-mer in the reference genome (excluding the SNV location and the first or last 10 kbp of repetitive sequence in the reference genome), we calculated the proportions of each non-reference base aligned to the middle nucleotide as a proxy for sequencing errors. Additionally, we calculated the proportion of alignments in which there was a deletion at the middle nucleotide. We then generated kernel density plots representing the distributions of these error proportions across all instances of the 5-mer in the reference genome (K3L$^{WT}$n = 126, K3L$^{His47Arg}$n = 83, E9L$^{WT}$n = 152, E9L$^{Glu495Gly}$n = 170), and calculated the median proportion of each sequencing error across all 5-mer sites.

## Generating simulated distributions of the K3L$^{His47Arg}$ variant within multicopy arrays

To simulate the accumulation of the K3L$^{His47Arg}$ allele in K3L arrays, we first created a simulated population of K3L arrays for each population that matched the copy number distribution of the passage of interest. Each copy within this population was initially K3L$^{WT}$; to simulate de novo accumulation of K3L$^{His47Arg}$ within these arrays, we looped over every copy within every K3L array. After randomly sampling a single value from a uniform distribution (0.0 to 1.0), if that value was less than or equal to the observed population allele frequency of K3L$^{His47Arg}$, we 'mutated' the selected copy. This process was repeated until the population allele frequency of the hypothetical population matched the observed population allele frequency at that passage. This process effectively simulated a binomial distribution of K3L$^{His47Arg}$ alleles within the hypothetical population, with the probability of mutating any particular K3L copy equal to the observed K3L$^{His47Arg}$ allele frequency in the passage.

## Simulating the effects of error rate on observations of mixed arrays

After empirically determining error rates for the 5-mers containing K3L$^{WT}$ or K3L$^{His47Arg}$, we simulated the effects of these error rates on our observed proportions of mixed and homogeneous arrays. To do this, we converted every vaccinia array in our experimental data into a homogeneous array using the following heuristic: if the array is already homogeneous, keep it as such; if the array contains a majority of K3L$^{WT}$ or K3L$^{His47Arg}$ alleles, convert it into an array of identical copy number that is homogeneous for the majority allele; if the array has an equal number of K3L$^{WT}$ and K3L$^{His47Arg}$ copies, randomly convert the array into a homogeneous K3L$^{His47Arg}$ or homogeneous K3L$^{WT}$ array of identical copy number. Then, for each K3L copy in each of these 'converted' arrays, we randomly introduced sequencing errors (i.e., switched K3L$^{His47Arg}$ alleles to K3L$^{WT}$ alleles, and vice versa) at a rate equivalent to the median C > T or T > C error rate for that sequencing chemistry.

## Accession numbers

All deep sequencing data are available on the Sequence Read Archive under accessions SRP128569 (Oxford Nanopore reads) and SRP128573 (Illumina MiSeq reads from the P15 and P20 populations). Previously published Illumina MiSeq reads from the P10 population are available on the SRA under accession SRP013146 (*Elde et al., 2012*). All Illumina MiSeq and Oxford Nanopore data are additionally archived on Zenodo at the following DOI: 10.5281/zenodo.1319732.

## Acknowledgements

We thank Ryan Layer for his assistance in early MinION sequencing experiments, and members of the Quinlan and Elde labs for helpful discussions during preparation of the manuscript.

## Additional information

### Competing interests

Thomas A Sasani: TAS received travel and accommodation expenses to speak at an Oxford Nanopore Technologies conference. The other authors declare that no competing interests exist.

### Funding

| Funder | Grant reference number | Author |
| --- | --- | --- |
| National Institutes of Health | R01GM114514 | Nels C Elde |
| Burroughs Wellcome Fund | 1015462 | Nels C Elde |
| University of Utah | Equipment Grant | Aaron R Quinlan<br>Nels C Elde |
| H.A and Edna Benning Presidential Endowed Chair | | Nels C Elde |
| National Institutes of Health | R01HG006693 | Aaron R Quinlan |

| National Institutes of Health | R01GM124355 | Aaron R Quinlan |
| National Institutes of Health | T32GM007464 | Thomas A Sasani |
| National Institutes of Health | T32AI055434 | Kelsey R Cone |

The funders had no role in study design, data collection and interpretation, or the decision to submit the work for publication.

## Author contributions

Thomas A Sasani, Software, Formal analysis, Validation, Investigation, Visualization, Methodology, Writing—original draft; Kelsey R Cone, Validation, Investigation, Visualization, Methodology, Writing—original draft; Aaron R Quinlan, Nels C Elde, Conceptualization, Supervision, Funding acquisition, Writing—review and editing

## Author ORCIDs

Thomas A Sasani (iD) https://orcid.org/0000-0003-2317-1374
Kelsey R Cone (iD) https://orcid.org/0000-0002-4547-7174
Aaron R Quinlan (iD) https://orcid.org/0000-0003-1756-0859
Nels C Elde (iD) http://orcid.org/0000-0002-0426-1377

## Decision letter and Author response

Decision letter https://doi.org/10.7554/eLife.35453.044
Author response https://doi.org/10.7554/eLife.35453.045

# Additional files

## Data availability

Sequencing data are publicly available at DOI: 10.5281/zenodo.1169394 Source data files are provided in the revised submission

The following dataset was generated:

| Author(s) | Year | Dataset title | Dataset URL | Database, license, and accessibility information |
| --- | --- | --- | --- | --- |
| Sasani TA, Cone KR, Quinlan AR, Elde NC | 2018 | Illumina MiSeq and Oxford Nanopore sequencing data from passaged dE3L vaccinia populations | http://dx.doi.org/10.5281/zenodo.1169394 | Creative Commons Attribution CC0, open access |

The following previously published dataset was used:

| Author(s) | Year | Dataset title | Dataset URL | Database, license, and accessibility information |
| --- | --- | --- | --- | --- |
| Elde NC, Child SJ, Eickbush, MT, Kitzman JO, Rogers KS, Shendure J, Geballe AP, Malik HS | 2012 | Illumina MiSeq sequencing data from the P10 dE3L vaccinia population. | https://www.ncbi.nlm.nih.gov/sra/?term=SRP013416 | Publicly available at the NCBI Sequence Read Archive (accession no. SRP013416) |

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
