## [Decision Letter]

[Editors’ note: the authors were asked to provide a plan for revisions before the editors issued a final decision. What follows is the editors’ letter requesting such plan.]

Thank you for sending your article entitled "Long read sequencing reveals poxvirus evolution through rapid homogenization of gene arrays" for peer review at *eLife*. Your article is being evaluated by three peer reviewers, and the evaluation is being overseen by Richard Neher as the Reviewing Editor and Patricia Wittkopp as the Senior Editor. The reviewers discussed your manuscript at length and one critical issue came up.

Sasani and colleagues use long read nanopore sequencing to characterize pox virus populations from cell culture evolution experiments. In these experiments, poxviruses adapt initially by amplification of the target gene or by a point mutation. The central claim of the paper is that the point mutation subsequently spreads among amplified gene arrays via a gene-conversion like mechanism. This is an intriguing possibility and long read sequencing seems in principle well suited to characterize this complex evolutionary dynamics. However, the evidence presented is also compatible with the following more parsimonious explanation:

The initial response of the virus population is gene amplification and the rise of single copy mutant genes. Once the latter are common, it becomes likely that they undergo gene amplification as well resulting in multi-copy mutant arrays. These homogeneous mutant arrays than gradually replace the WT arrays. This mechanism explains the homogeneity of the arrays and their late dominance via processes we already know to be common in this system without the need to invoke gene conversion. We failed to identify evidence in the data you presented that favors gene conversion over this simpler mechanism, especially given the high error rate of nanopore sequencing.

Given this critical issue, the editors and reviewers invite you to respond within the next two weeks either with existing data/explanations in favor of gene conversion, or suggestions for experiments/analyses that would decisively answer this question. We plan to share your responses with the reviewers and then issue a binding recommendation.

[Editors’ note: formal revisions were requested, following approval of the authors’ plan of action.]

Thank you for submitting your article "Long read sequencing reveals poxvirus evolution through rapid homogenization of gene arrays" for consideration by *eLife*. Your article has been reviewed by three peer reviewers, including Richard A Neher as the Reviewing Editor and Reviewer #1, and the evaluation has been overseen by Patricia Wittkopp as the Senior Editor.

The reviewers have discussed the reviews with one another and the Reviewing Editor has drafted this decision to help you prepare a revised submission.

Summary:

Sasani and colleagues use long read nanopore sequencing to characterize pox virus populations from cell culture evolution experiments. The central claim of the paper is that pox virus achieves immune evasion initially by either amplification of the target gene or a point mutation. Subsequently, the point mutation spreads among amplified gene arrays via a gene-conversion like mechanism. This is an intriguing possibility and long read sequencing is well suited to characterize these complex evolutionary dynamics.

Essential revisions:

Sasani et al., have already responded to our most serious concern, an alternative mechanism via amplification of mutant alleles, and their additional analyses provide convincing evidence that this simple mechanism can be ruled out. We would like these analyses to be included in the manuscript and have a number of additional essential points that need to be addressed:

1) Statistics on the arrangements of alleles in multi-copy arrays. How often are different patterns AAAAA, AABAA, ABBAB, etc. observed? Do identical alleles tend to be neighbors or are they randomly distributed? Are specific subpatterns over-represented across 3,4,5 copy arrays? These statistics should be used to explicitly discuss the plausibility of different scenarios and ideally quantitatively model the dynamics across time. Simply showing examples/pictures and stating that this result is "compatible" with your hypothesis is not enough. For follow-up analysis, it would be useful to provide alignments of 1, 2, 3, 4, 5, copy arrays such that readers don't have to go back to the FASTQ files (maybe after stripping non-reference insertions that are likely ONT errors).

2) Recombination and MOI: Viral titer will increase dramatically during the 48h experiments and there is, therefore, no single MOI for any experiment. Furthermore, even experiments started at low MOI likely have high MOI towards the end. Subsection “Recombination and selection drive patterns of K3L^His47Arg^ homogenization” should explain these caveats and discuss carefully the degree to which inter-genomic recombination can be ruled out. Statistical signatures of WT/mutant patterns in the multicopy arrays might help to differentiate such patterns (cross-over recombination might result in mutant copies being preferentially at the 3' or 5' end of the array).

Qin and Evans, 2014 provide careful estimates of pox virus recombination rates. In particular, they find short recombination tract length (that look like gene conversion) and frequent recombination even though experiments start at MOI 0.02. This further calls into question your assumption that little inter-genomic recombination is happening.

If inter-genomic recombination is common, the observed patterns could be explained by non-allelic homologous recombination that results in frequent deletion, concatentation, and replacements of arrays or parts thereof.

3) ONT error rates. A detailed analysis of sequencing errors needs to be part of the manuscript. It would be useful to indicate the ONT error threshold in Figure Figure 2B. How does one reconcile the fact that ONT finds 10% mutant alleles in P5 while these are not detected in Illumina reads?

4) An effort should be made to clarify the logic of the manuscript. The reviews below highlight problems and provide a number of concrete suggestions to improve the manuscript.

5) "Homology" is used incorrectly: homology is a binary quality. Two sequences either share a common ancestor (are homologous) or they don't. You use homology as a synonym for similarity, which it is not (see https://www.ncbi.nlm.nih.gov/books/NBK20255/).

6) Better graphs: bar graphs are a sub-optimal way to present data in many cases. In Figure 1A/D, for example, please show all data points and the median, there is no need for the bars. Figure Figure 1—figure supplement 1 is even worse - all the relevant differences happen within 10% of the figure. Figure 2C might be better as line graph with copy number as x-axis and one line per passage. We would like to see graphs and statistics that explicitly show the WT/mutant arrangements in multi-copy arrays with more information than the current Figure 3 and Figure 4. Overall, quantitative analysis and presentation of the data should be improved.

Finally, you need to be careful not to overstate your case: Evidence for gene conversion remains indirect (experiments with mixtures of viruses recoded at synonymous positions might provide more direct evidence), the importance of this process in the real-world pox-virus infections remains speculative, and other mechanisms to homogenize the array can't be ruled out.

---

## [Author Response]

[Editors’ note: what follows is the authors’ plan to address the revisions.]

Sasani and colleagues use long read nanopore sequencing to characterize pox virus populations from cell culture evolution experiments. In these experiments, poxviruses adapt initially by amplification of the target gene or by a point mutation. The central claim of the paper is that the point mutation subsequently spreads among amplified gene arrays via a gene-conversion like mechanism. This is an intriguing possibility and long read sequencing seems in principle well suited to characterize this complex evolutionary dynamics. However, the evidence presented is also compatible with the following more parsimonious explanation:The initial response of the virus population is gene amplification and the rise of single copy mutant genes. Once the latter are common, it becomes likely that they undergo gene amplification as well resulting in multi-copy mutant arrays. These homogeneous mutant arrays than gradually replace the WT arrays. This mechanism explains the homogeneity of the arrays and their late dominance via processes we already know to be common in this system without the need to invoke gene conversion. We failed to identify evidence in the data you presented that favors gene conversion over this simpler mechanism, especially given the high error rate of nanopore sequencing.Given this critical issue, the editors and reviewers invite you to respond within the next two weeks either with existing data/explanations in favor of gene conversion, or suggestions for experiments/analyses that would decisively answer this question. We plan to share your responses with the reviewers and then issue a binding recommendation.

We appreciate the concern and outline here how the existing data support our proposed mechanism of gene conversion. We also took this opportunity to perform some new analyses. Taking the existing data and new analyses, we highlight three lines of evidence supporting a model of gene conversion. Although some homogeneous replacement of wild type K3L arrays likely occurs, we think you will agree that a closer examination of the data strongly supports a primary role for recombination-driven homogenization of the K3L arrays, revealing the importance of gene conversion for poxvirus adaptation. We feel that this remains a novel and important finding with potentially far reaching implications, particularly for large DNA virus evolution, meriting further consideration for publication in *eLife*.

Here we outline the major points supporting our model, with details of our new analyses. We expect that this and related modifications would be included in a revised manuscript:

The prevalence and complexity of the mixed (mutant and WT K3L) multicopy genomes we observe favors a recombination-based mechanism for variant homogenization. We agree that the current presentation of data, for example in Figure 3 and Figure 4, makes it difficult to distinguish the mechanisms underlying our conclusions. However, random sampling and genotyping of vaccinia genomes (Author response image 1), strongly supports the existence and variety of mixed arrays, even when accounting for error rates in nanopore sequencing, which we further analyzed in the next point.

**Author response image 1. respfig1:** Variety of mixed vaccinia genomes in P15 sequencing data. We selected a random set of 20 vaccinia genomes of the specified copy number from the P15 sequencing data (a passage with a large proportion of mixed genomes), and plotted the distribution of K3L^His47Arg^ and K3L^WT^ alleles within each genome.

We calculated the sequencing error rate for the exact sequence change encoding the K3L^His47Arg^ variant using our nanopore data. We demonstrate that the relatively low error rate for calling this mutation (<3%) cannot account for the abundance of mixed genomes (see Point 2). Furthermore, the proportion of mixed genomes we observe is consistent across multiple nanopore sequencing chemistries (3 versions over the course of our study) and is therefore robust to changes in error rate (see Point 2, Figure 2 and Table 1). This new analysis also includes simulations of K3L array variant calling given the nanopore error rates, which further support the veracity of our original interpretation of the data (Point 2, Figure 3 and Table 2).

Independent evidence for mixed array homogenization emerged from our observations during the course of the experiments. Specifically, the K3L^His47Arg^ variant is present at a consistent frequency of roughly 10% from P5 to P10, such that the substantial expansion of multicopy genomes that occurs during these passages are almost entirely K3L^WT^ (see Figure 1B of original manuscript). As the expansion of K3L arrays reaches a plateau by P10, the frequency of mixed genomes rapidly increases, with K3L^His47Arg^ reaching nearly 90% frequency by P20. We simultaneously tracked a nearby variant, a 3 bp shift at the site of the recombination breakpoints. The resulting two breakpoint variants appear to be neutral and, in sharp contrast to K3L^His47Arg^, they remain at nearly identical frequencies from P5 to P20 of the experiments (see Figure 6D and Table S1 in the original manuscript). Our interpretation of these data is that virus populations “top out” in K3L^WT^ copy number before the His47Arg mutation begins infiltrating K3L arrays. Under the alternate model proposed in review, we would predict that one of the two neutral breakpoints would preferentially ‘hitchhike’ in homogeneous K3L^His47Arg^ arrays, and at the very least change substantially in frequency. More consistent with these data, a model of recombination-driven homogenization can account for the consistency of the neutral variants at a nearby genomic location (a point we propose to clarify and emphasize in a revised manuscript).

We anticipate including these new analyses and also sharpening the presentation of evidence for gene conversion in a revised version of the manuscript. Please find below details supporting and clarifying our claims:

Point 1: The large proportion of mixed alleles is consistent with recombination-based accumulation and homogenization of the K3L^His47Arg^ variant.

As the K3L^His47Arg^ variant accumulated in the population, the high proportion of multicopy genomes containing mixed sets of alleles suggests that rampant recombination occurred, in line with a model involving gene conversion (subsection “The K3L^His47Arg^ variant rapidly homogenizes in multicopy vaccinia genomes”). For example, these mixed K3L gene arrays comprise nearly ¼ of all vaccinia genomes sequenced at P15 (Figure 3, original submission). Given the high recombination rate and relatively low point mutation rate of poxviruses, these mixed arrays are likely a product of recombination events rather than repeated de novo point mutations.

The reviewers raise the point that the mixed arrays could also be a result of the higher error rate of nanopore sequencing (which we further address below). Were this the case, we would expect the vast majority of these mixed genomes to be homogeneous except for one K3L copy containing a different allele. However, we observed a large variety of mixed K3L^WT^ and K3L^His47Arg^ copies in multicopy genomes (Author response image 1).

Furthermore, we observed various combinations of K3L alleles in mixed genomes, rather than alleles of each type consistently grouped together (Author response image 1). These data are indicative of successive (rather than single) recombination events, further supporting a recombination-based mechanism of variant accumulation and homogenization. Copy number expansions from a single-copy K3L^His47Arg^ genome can, and likely do, occur, but the abundance of mixed arrays suggests that this is not the primary mechanism of homogenization, as proposed under the alternative model. The presence of the mixed arrays better fits a model of recombination spreading the variant through multicopy genomes.

Point 2: Nanopore sequencing error rates at the K3L^His47Arg^ locus are low, and do not explain the large fraction of mixed genomes we observe.

One possible concern is that the high error rate of nanopore sequencing may be impacting our observations of homogeneous and mixed arrays. Such a concern understandably could stem from our lack of clarity regarding the sources and rates of nanopore sequencing error in our data as originally presented. Because nanopore sequencing involves the measurement of changes in ionic current caused by specific *k*mers occupying the nanopore, rather than individual nucleotides, sequencing accuracy varies depending on the particular *k*-mer being sequenced. Indeed, counts of homopolymer *k*-mers are often underrepresented in ONT reads, when compared to their counts in the reference genome of the sequenced sample (Loman et al., 2015).

Therefore, we have more clearly defined the error rates in our sequencing data. To do this, we calculated the proportions of sequencing errors (mismatches and deletions) at all other 5-mers in the vaccinia genome that match the two 5-mer sequences that include the nucleotide encoding either K3L^WT^ or K3L^His47Arg^ (WT: TATGC, *n*=126, His47Arg: TACGC, *n*=83; see Extended Methods below). Using three separate nanopore sequencing chemistries, we find that T>C and C>T errors at these 209 loci are slightly more common than other single nucleotide errors; however, the median proportions of these errors are never greater than 2.6% (Author response image 2 and Response table 1). In stark contrast, using these R7.3, R9, and R9.4 sequencing chemistries to sequence P15, we estimate the population allele frequency of K3L^His47Arg^ (encoded by a T>C variant) in multicopy genomes to be 53.8%, 55.0%, and 57.0%, respectively. These observed allele frequencies at the K3L locus are clearly well above the basal ONT error rates for these 5-mers, and demonstrate that sequencing error cannot account for the prevalence and complexity of mixed genomes we observe.

**Author response image 2. respfig2:** Error rate distribution within K3L^His47Arg^ sequence context. (**A**) Kernel density plots representing the distribution of error rates for T>C, T>A, T>G, and T>deletion error across all TATGC 5-mers in data from each flowcell chemistry. (**B**) Kernel density plots representing the distribution of error rates for C>T, C>A, C>G, and C>deletion errors across all TACGC 5-mers in data from each flowcell chemistry.

Response table 1: Median T>C and C>T sequencing error rates using various ONT chemistries

R7.3 ONT chemistryR9 ONT chemistryR9.4 ONT chemistryT > C0.0230.0230.005C > T0.0150.0240.026

Although we used different flow cell chemistries throughout the course of our experiments, and each chemistry has a different error profile, we observed a nearly identical proportion of mixed genomes when sequencing the same population with each of the different flow cell chemistries (Author response image 3). This suggests that our observations of mixed arrays are robust to both flow cell chemistry and sequencing error rates. Our calculations of error for the specific 5-mer surrounding the K3L^His47Arg^ allele show that the later flow cell chemistries have little T>C bias (Author response image 2), and yet we still observed a high proportion of mixed reads using these flow cells. As mentioned above, if rare sequencing errors were driving our observations of mixed genomes, we might expect that the most common types of mixed genomes in our data would be “near homogeneous” (i.e., homogeneous except for one copy due to a sequencing error). Therefore, we calculated binomial p-values to assess whether we observe more of these “near homogeneous” genomes than expected given the T>C and C>T error rates in our data (Extended Methods). For all flow cell chemistries, these pvalues are extremely small (Response Table 2), suggesting that the mixed genomes we observe are truly heterogeneous, rather than artifacts.

**Author response image 3. respfig3:** Oxford Nanopore sequencing error does not impact observed patterns of K3L copy number or K3L^His47Arg^ allele heterogeneity. The P15 vaccinia population was sequenced with R7.3, R9, and R9.4 chemistry ONT flowcells. (**A**) Stacked bar plots representing the diversity of allele combinations within single-copy and multicopy reads were generated from sequenced ONT reads from each chemistry, as in Figure 3 and Figure 5 (original manuscript). (**B**) All mixed genomes observed in the data from each flowcell chemistry were converted to homogeneous genomes (Extended Methods), and sequencing errors were randomly distributed throughout these genomes at a rate equivalent to the median C>T or T>C error rate for the particular chemistry (see Response Table 1). Stacked bar plots representing the resulting diversity of allele combinations were then created as in (**A**). (**C**) We performed the simulation in (**B**) a total of 1000 times, and generated kernel density plots representing the distribution of mixed genome proportions recovered in each simulation. The red line indicates the proportion of mixed genomes observed in the experimental data for the sequencing chemistry of interest..

Response Table 2: Binomial p-values calculated for observed numbers of near-homogeneous (NH) mixed genomes

R7.3 ONT chemistryR9 ONT chemistryR9.4 ONT chemistryP *(NH WT)*4.46E-53.21E-382.48E-42P *(NH H47R)*2.86E-201.54E-2356.03E-123

Furthermore, if we assume that all genomes in our data are truly homogeneous and then randomly distribute T>C and C>T errors throughout these genomes at a frequency equivalent to their median error rates for each chemistry, we find that these errors do not account for the high proportion of mixed arrays that we observe in our experimental data (Author response image 3). We repeated this experiment (i.e., randomly distributing sequencing errors) a total of 1,000 times, and find that our observed proportion of mixed genomes is always greater than the results of simulations (Author response image 3). Using R7.3, R9, and R9.4 chemistry flowcells, the observed rate was roughly five times the rate expected from sequencing error (defined as the median of 1,000 simulations). Thus, while the error rate of nanopore sequencing is higher than that of technologies like Illumina, it does not have a large impact on our data. These internal controls further support our accurate identification of mixed genomes, suggesting that recombination is rampant, and thus the spread of the K3L^His47Arg^ variant cannot be due to expansion of homogeneous genomes alone.

Point 3: Multicopy homogeneous K3L^His47Arg^ genomes do not initially accumulate, despite abundant single-copy K3L^His47Arg^ genomes.

Under a model of homogeneous array replacement, single-copy K3L^His47Arg^ genomes undergo gene amplification once they become common in the population, and subsequently replace WT homogeneous genomes. However, this model does not match our observations. As early as P5, the K3L^His47Arg^ variant was present in nearly 40% of single-copy genomes. At P10, however, we observed very few multicopy genomes homogeneous for the K3L^His47Arg^ allele (original manuscript, Figure 3). Were selection for homogeneous K3L^His47Arg^ genomes the main driver of variant accumulation, we would expect to see a larger fraction of these genomes at P10. Furthermore, although overall K3L copy number increases from P5 to P10, this pattern is driven almost entirely by the amplification of homogeneous K3L^WT^ alleles.

Across the same time points (P5-10), the K3L^His47Arg^ allele remains at a constant population-level frequency (original manuscript, Figure 2B, Figure 6—source data 1), but there is a shift in the proportion of multicopy genomes that contain K3L^His47Arg^ alleles. At P5, K3L^His47Arg^ is present almost exclusively in single-copy genomes, but by P10 the allele has spread to a larger proportion of multicopy genomes, which are mainly mixed rather than homogeneous (original manuscript, Figure 3). These results suggest that the K3L^His47Arg^ allele was present in a large proportion of single-copy genomes for many passages, but did not undergo significant gene amplification as predicted by the reviewer's model. Instead, we observed the rapid accumulation and homogenization of the K3L^His47Arg^ allele only from P10 to P20, accompanied by a large population of mixed arrays (original manuscript, Figure 3). This suggests that the variant is introduced into multicopy arrays through recombination, and only once these mixed genomes are more abundant did we observe an increase in homogeneous K3L^His47Arg^ multicopy genomes. Gene amplification of single-copy K3L^His47Arg^ genomes is of course possible, and likely occurs as well, but does not seem to be the main mechanism of array homogenization.

Coupled with our observations of consistent recombination breakpoint variant frequencies (original manuscript, Figure 6D and Table S1), these data favor a recombination-driven mechanism of adaptation.

Extended Methods:

Calculating sequence-dependent error rates in nanopore sequencing data.

We scanned the vaccinia reference genome for every instance of the “TATGC” 5-mer sequence; the thymine at the 3rd position of this sequence (at reference position 30,490) is the mutated base that encodes the K3L^His47Arg^ amino acid change. For each instance of this 5-mer in the reference genome (ignoring the 5-mer that spans reference positions 30,388-30,492), we calculated the proportions of As, Cs, and Gs aligned to the site of the middle nucleotide in the reference genome; observations of these non-reference bases likely represent sequencing errors. Additionally, we calculated the proportion of alignments in which there was a deletion at the middle nucleotide. We then generated kernel density plots representing the distributions of these error proportions across all instances of the 5-mer in the reference genome (*n* = 126, excluding any *k*mer matches in the first or last 10 kbp of the vaccinia genome, which comprise highly repetitive sequence). We repeated this analysis for all “TACGC” 5-mer sequences (*n =* 83, excluding sequence in the first or last 10 kbp of the genome); a cytosine to thymine change at the center nucleotide would encode a K3L^His47Arg^ to K3L^WT^ change, and is therefore a proxy for sequencing errors that would lead us to incorrectly call a K3L^His47Arg^ copy as being K3L^WT^.

Simulating the effects of error rate on observations of mixed and homogeneous genomes.

We then simulated the effects of the empirically determined T>C and C>T error rates on our observed proportions of mixed and homogeneous genomes. In other words, we addressed the following question: if all of the vaccinia genomes in our passaged populations were, in fact, homogeneous for K3L^WT^ or K3L^His47Arg^ alleles, would we expect to observe a large fraction of mixed genomes simply due to sequencing errors? To do this, we converted every vaccinia genome in our experimental data into a homogeneous array using the following heuristic: if the genome is already homogeneous, keep it as such; if the genome contains a majority of K3L^WT^ or K3L^His47Arg^ alleles, convert it into a genome of identical copy number that is homogeneous for the majority allele; if the genome has an equal number of K3L^WT^ and K3L^His47Arg^ copies, randomly convert the genome into a homogeneous K3L^His47Arg^ or homogeneous K3L^WT^ genome of identical copy number. Then, for each K3L copy in each of these “converted” genomes, we randomly introduced sequencing errors (i.e., switched K3L^His47Arg^ alleles to K3L^WT^ alleles, and vice versa) at a rate equivalent to the median C>T or T>C error rate for that sequencing chemistry.

If we perform this simulation using the median error rates for each flow cell chemistry, we observe far fewer mixed genomes than in our experimental data, suggesting that the large proportions of mixed genomes in Figure 3 (main manuscript) represent true heterogeneous K3L arrays (Author response image 3). To address the statistical significance of this observation, we performed two experiments.

First, we performed the simulation outlined above a total of 1000 times. For each simulation, we calculated the proportion of genomes with mixed alleles, and plotted the distributions of mixed genome proportions across all 1000 simulations (Author response image Figure 3C). For all sequencing chemistries, we find that our observed proportion of mixed genomes greatly exceeds (~5 times) the expected proportion of mixed genomes recovered by simulations (Author response image 3).

Second, for all genomes of copy number two or greater, we calculated the proportion of genomes that were homogeneous for either K3L^WT^ or K3L^His47Arg^ alleles, as well as the proportion of genomes that were “near-homogeneous” (containing all but one K3L^WT^ or K3L^His47Arg^ allele). To test if we observe more of these “near-homogeneous” mixed arrays than expected given our C>T and T>C error rates, we calculated a binomial pvalue as follows. In the following examples, “near-homogeneous WT genomes” are those with all but one K3L^WT^ copy, and “near-homogeneous H47R genomes” are those with all but one K3L^His47Arg^ copy.

WTNH = # of near-homogeneous WT genomes

WTH = # of homogeneous WT genomes

H47RNH = # of near-homogeneous H47R genomes

H47RH = # of homogeneous H47R genomes

ECT = median C>T error rate for the flowcell chemistry

ETC = median T>C error rate for the flowcell chemistry

P*(NH WT)* = binom(WT_NH_, WT_NH_ + W_TH_, probability = E_TC_) = probability of observing WT_NH_ given E_TC_

P*(NH H47R)* = binom(H47R_NH_, H47R_NH_ + H47_RH_, probability = E_CT_) = probability of observing H47R_NH_ given E_CT_

For all genomes with 2 or more copies of K3L, the p-values of each binomial test are presented in Response Table 2.

[Editors’ notes: the authors’ response after being formally invited to submit a revised submission follows.]

Summary:Sasani and colleagues use long read nanopore sequencing to characterize pox virus populations from cell culture evolution experiments. The central claim of the paper is that pox virus achieves immune evasion initially by either amplification of the target gene or a point mutation. Subsequently, the point mutation spreads among amplified gene arrays via a gene-conversion like mechanism. This is an intriguing possibility and long read sequencing is well suited to characterize these complex evolutionary dynamics.Essential revisions:Sasani et al.et al., have already responded to our most serious concern, an alternative mechanism via amplification of mutant alleles, and their additional analyses provide convincing evidence that this simple mechanism can be ruled out. We would like these analyses to be included in the manuscript and have a number of additional essential points that need to be addressed:

In response to this previous concern, we performed an extensive analysis of error rates and related issues, which we sent as a response and supporting PDF before receiving full reviews. The pertinent points of this analysis are now incorporated as follows into the revised manuscript (note: all line numbers included below refer to the final revised version of the manuscript and may differ from those in the original document).

1) Additional analyses and explanations of ONT error rates. See Figure 2—figure supplement—figure supplement 2, Figure 3—figure supplement 2, Figure 3—figure supplement 3, Table 2 descriptions in the manuscript (subsection “The K3L^His47Arg^ variant rapidly homogenizes in multicopy vaccinia genomes”, subsection “Deep sequencing of viral genomes”), and an updated Materials and methods section.

2) More careful consideration of the combined data supporting a model resembling gene conversion rather than expansion of homogeneous K3L^His47Arg^ arrays in Figure 3—figure supplement 4, Figure 4B, descriptions in the manuscript (subsection “Copy number and allele frequency estimation” and subsection “Breakpoint characterization”), and a revised Discussion section.

1) Statistics on the arrangements of alleles in multi-copy arrays. How often are different patterns AAAAA, AABAA, ABBAB, etc. observed? Do identical alleles tend to be neighbors or are they randomly distributed? Are specific subpatterns over-represented across 3,4,5 copy arrays? These statistics should be used to explicitly discuss the plausibility of different scenarios and ideally quantitatively model the dynamics across time. Simply showing examples/pictures and stating that this result is "compatible" with your hypothesis is not enough. For follow-up analysis, it would be useful to provide alignments of 1, 2, 3, 4, 5, copy arrays such that readers don't have to go back to the FASTQ files (maybe after stripping non-reference insertions that are likely ONT errors).

We have added a supplement to Figure 3 (Figure 3—figure supplement 4) that shows a more comprehensive analysis of 3-copy K3L arrays containing every possible combination of K3L^WT^ and K3L^His47Arg^ alleles in passages 10, 15, and 20 (subsection “Copy number and allele frequency estimation”). We chose to focus on combinations in 3-copy arrays for two main reasons: first, because 3-copy arrays make up a large fraction of multi-copy K3L arrays in these passages (and represent a useful cross-section of the diversity of multi-copy arrays); and second, because our sequencing datasets for passages 10 and 20 did not contain enough reads to create robust and meaningful plots of 4- and 5-copy array combinations. In addition, we performed additional deep sequencing to profile virus genomes at passage 15 (which contained the largest fraction of mixed arrays among any of our passaged datasets), generating an additional ~2,700 reads that fully aligned to the K3L duplicon. Using these additional data from P15, we generated plots of mixed array combinations for 4- and 5-copy arrays. Although the technical limitations of our sequencing approach prevent us from generating large numbers of reads containing 6+ K3L copies, we expect that the trends we observe in 3, 4, and 5-copy arrays hold for higher copy-number arrays as well, based on our current dataset for robustly scoring large arrays.

We also considered a number of possible statistical analyses to address whether particular allele combinations were over- or under-represented in mixed K3L arrays. Ultimately, though, we struggled to build a null expectation of K3L^His47Arg^ accumulation to which we could compare our experimental data. For example, to simulate the accumulation of K3L^His47Arg^ through de novomutation alone, we considered uniformly distributing K3L^His47Arg^ alleles throughout a population that initially contained only K3L^WT^ alleles, and subsequently counting the numbers of arrays with each combination of K3L^WT^ and K3L^His47Arg^ alleles. However, given the low point mutation rates of poxviruses, this seemed like a highly unrealistic model of K3L^His47Arg^ accumulation. Additionally, we considered a model in which we grouped all K3L arrays of a given copy number at P10, P15, and P20 by the number of crossovers needed to generate them (an “AAAA” or “BBBB” array would require zero crossovers, an “ABBB” or “BBBA” array would require one, etc.). Then, we calculated the average number of crossovers we observed among all K3L arrays in a passage, and estimated the number of arrays with 0, 1, … *n* crossovers we would expect based upon a Poisson distribution. Although this model is relatively simple, it suffers from the requirement that we build an expectation from our observed experimental data. This inherent circularity is troubling and therefore, we ultimately decided to present our analyses of K3L allele combinations in multicopy K3L arrays without explicit statistical treatment. We believe that the trends shown in Figure 3—figure supplement 4 clearly demonstrate that the allelic patterns observed in mixed arrays are distinct between passages, sum to a significant fraction of all arrays, and support our proposed mechanism of genetic homogenization.

Additionally, we have added a panel to Figure 4 in which we excluded all homogeneous arrays from the P15 dataset, and plotted the frequency of K3L^His47Arg^ in each copy of the remaining mixed K3L arrays. We show that K3L^His47Arg^ homogenizes within multicopy K3L arrays even when we exclusively examine mixed arrays, and demonstrate that K3L^His47Arg^ accumulation is independent of K3L array copy number. This new Figure 4B and modified text (subsection “Breakpoint characterization”) provides strong evidence of K3L homogenization through a recombination-based mechanism, rather than amplification of purely K3L^WT^ or K3L^His47Arg^ arrays.

Finally, for ease of future analysis, we have uploaded aligned sequencing reads (in BAM format) for all analyzed passages to the SRA (accession SRP128569) and Zenodo (DOI 10.5281/zenodo.1319732). These alignment files contain only those reads that aligned to the K3L locus.

2) Recombination and MOI: Viral titer will increase dramatically during the 48h experiments and there is, therefore, no single MOI for any experiment. Furthermore, even experiments started at low MOI likely have high MOI towards the end. Subsection “Recombination and selection drive patterns of K3LHis47Arg homogenization” should explain these caveats and discuss carefully the degree to which inter-genomic recombination can be ruled out. Statistical signatures of WT/mutant patterns in the multicopy arrays might help to differentiate such patterns (cross-over recombination might result in mutant copies being preferentially at the 3' or 5' end of the array).Qin and Evans, 2014 provide careful estimates of pox virus recombination rates. In particular, they find short recombination tract length (that look like gene conversion) and frequent recombination even though experiments start at MOI 0.02. This further calls into question your assumption that little inter-genomic recombination is happening.If inter-genomic recombination is common, the observed patterns could be explained by non-allelic homologous recombination that results in frequent deletion, concatentation, and replacements of arrays or parts thereof.

We agree that multiple rounds of replication influence MOI and increase the chances of intergenomic recombination during the experiments. However, in specific consideration of results from Qin and Evans, (2014), our experimental approach differs in three notable ways. First, our viruses have reduced replication ability compared to WT viruses; therefore, the potential for increases in MOI over the course of 48 hours is diminished. Second, our lowest MOI is 20-fold lower than reported in previous work, which reduces the occurrence of co-infection and intergenomic recombination in our protocol. A quick estimation of the maximum titer following a 48-hour passage in our MOI 0.001 experiment predicts 2-3 infectious particles per cell, which further supports the rarity of co-infections in our setup by the time virus is harvested. Finally, the previous observations of short recombination tract length (Qin and Evans, 2014), which are also supported by Fathi et al.et al., 1991, are only seen in high MOI (10) single passage experiments.

Therefore, while intergenomic recombination seems to favor short conversion tracts, the more modest potential for co-infection in our MOI 0.001 experiment suggests that this is not the main mechanism by which the SNV accumulates. Given the similar proportions of mixed and homogeneous K3L arrays we observed across a 10,000-fold difference in MOI (Figure 5), the data support the idea that intergenomic recombination does not play a significant role in the homogenization of K3L^His47Arg^. We have more clearly acknowledged these important caveats and explain these points more clearly in the revised manuscript (subsection “Generating simulated distributions of the K3LHis47Arg variant within multicopy arrays”).

3) ONT error rates. A detailed analysis of sequencing errors needs to be part of the manuscript. It would be useful to indicate the ONT error threshold in Figure 2B. How does one reconcile the fact that ONT finds 10% mutant alleles in P5 while these are not detected in Illumina reads?

As described in our previous response to the reviewers’ major concern noted above, we performed additional analyses to more clearly explain and calculate the error rates in our ONT data. This is described in subsection “The K3L^His47Arg^ variant rapidly homogenizes in multicopy vaccinia genomes”, subsection “Deep sequencing of viral genomes”, as well as in the Materials and methods section. To illustrate these error rates, we have added supplements for Figure 2 and Figure 3 (Figure 2—figure supplement 2, Figure 3—figure supplement 2, and Figure 3—figure supplement 3).

We have chosen not to annotate Figure 2B with ONT error thresholds, as the methods we used to generate allele frequency estimates differ from the methods used to calculate error rates. Specifically, our estimates of ONT error rate were calculated directly from read alignments, by counting every nonreference base aligned to the middle nucleotide of the 5-mer of interest. ONT allele frequencies, on the other hand, were estimated using the nanopolish software tool. Nanopolish examines the aligned sequences in a BAM file, as well as the raw current signal output by the MinION device, to increase the accuracy of SNV calling. Therefore, we do not feel that the two measurements are directly comparable. However, we have included our measurements of error rate for each sequencing chemistry and SNV in a supplement to Figure 2 (Figure 2—figure supplement 2), and in an updated version of Table 2, so that readers are made aware of the error rates in the manuscript. Also, in regard to Figure 2B, we did not sequence P5 using Illumina, as described in the updated figure legend. Thus, it is not the case that the mutant allele was not detected in Illumina reads at P5. We simply do not have Illumina data for that passage and apologize for the confusion. For clarity, we have updated text annotations in the figure (Figure 2B) and its legend from “n.d.” (not determined – but potentially interpreted as not detected) to “n.t.” (not tested).

4) An effort should be made to clarify the logic of the manuscript. The reviews below highlight problems and provide a number of concrete suggestions to improve the manuscript.

We agree and have reorganized the first three sections of the manuscript to improve the logical progression and grouping of ideas and experiments. First, we moved the description of the only other high-frequency SNV (E9L^Glu495Gly^) into the first section of the manuscript (subsection “ONT reads reveal precise K3L copy number and distributions of K3L^His47Arg^ in individual genomes”) to clarify our reasoning for focusing on genetic changes at the K3L locus (see also response to reviewer #3’s last comment). Second, we combined the second and third sections of the manuscript under a new heading (subsection “The K3L^His47Arg^ variant rapidly homogenizes in multicopy vaccinia genomes”) to better organize and emphasize the information obtained from our initial ONT sequencing experiments (see also response to reviewer #3’s first comment).

5) "Homology" is used incorrectly: homology is a binary quality. Two sequences either share a common ancestor (are homologous) or they don't. You use homology as a synonym for similarity, which it is not (see https://www.ncbi.nlm.nih.gov/books/NBK20255/).

We agree and have corrected the manuscript accordingly (Introduction). We maintain the use of homology/homologous in instances where referring to sequences that share a common ancestor between species, and also for sequences within a gene family that share a common ancestor within the genome.

6) Better graphs: bar graphs are a sub-optimal way to present data in many cases. In Figure 1A/D, for example, please show all data points and the median, there is no need for the bars. Figure 1—figure supplement 1 is even worse -- all the relevant differences happen within 10% of the figure. Figure 2C might be better as line graph with copy number as x-axis and one line per passage. We would like to see graphs and statistics that explicitly show the WT/mutant arrangements in multi-copy arrays with more information than the current Figure 3 and Figure 4. Overall, quantitative analysis and presentation of the data should be improved.

We have updated Figure 1A, Figure 1D, Figure 1—figure supplement 1, and Figure 1—figure supplement 3 to more clearly represent the data. These have been changed from bar graphs to dot plots showing each data point, median, and 95% CI. As stated above in response to the first point, we have included additional analyses of the variant patterns within multicopy arrays in Figure 3—figure supplement 4. We have also included an example of Figure 2C redrawn as a line plot below (Author response image 4); however, we feel that the current figure design, using stacked bars to represent the proportions of arrays with a given copy number, is more useful.

**Author response image 4. respfig4:** 

Finally, you need to be careful not to overstate your case: Evidence for gene conversion remains indirect (experiments with mixtures of viruses recoded at synonymous positions might provide more direct evidence), the importance of this process in the real-world pox-virus infections remains speculative, and other mechanisms to homogenize the array can't be ruled out.

We have updated the Discussion section to more clearly acknowledge other potential mechanisms and better summarize the evidence from our experiments. While we think tracking the noncoding variation in breakpoints near K3L (Figure 2A) offer some evidence, we agree that a synonymous site re-coding and mixing experiment would be more incisive and thank the reviewer for making the suggestion.